# SLQ: Bridging Modalities via Shared Latent Queries for Retrieval with Frozen MLLMs

**Haoran Lou** [1]   **Ziyan Liu** [1]   **Chunxiao Fan** [1]   **Yuexin Wu** [1]   **Yue Ming** [1]
**Hao Wu** [2]   **Kai Zuo** [2]   **Yibo Chen** [2]   **Xu Tang** [2]

## Abstract

Multimodal Large Language Models (MLLMs) possess intrinsic reasoning and world-knowledge capabilities, yet adapting them for dense retrieval remains challenging. Existing approaches rely on invasive parameter updates, such as full fine-tuning and LoRA, which may disrupt the pre-trained semantic space and impair the structured knowledge essential for reasoning. To address this, we propose **SLQ**, a parameter-efficient tuning framework that adapts MLLMs for retrieval while keeping the backbone entirely frozen. SLQ introduces a small set of **Shared Latent Queries** that are appended to both text and image tokens, leveraging the model's native causal attention to aggregate multimodal context into a unified embedding space. Furthermore, to better evaluate retrieval beyond superficial pattern matching, we construct **KARR-Bench**, a benchmark designed for knowledge-aware reasoning retrieval. Extensive experiments show that SLQ outperforms full fine-tuning and LoRA on COCO and Flickr30K, while achieving competitive performance on MMEB and yielding substantial gains on KARR-Bench, validating that preserving the pre-trained representations via non-invasive adaptation is an effective strategy for MLLM-based retrieval. The code is available under: https://github.com/CnFaker/SLQ.

## 1. Introduction

Multimodal Large Language Models (MLLMs) (Liu et al., 2024a; An et al., 2025; Lou et al., 2025; Bai et al., 2025b; Wang et al., 2025; Team et al., 2024; Achiam et al., 2023)

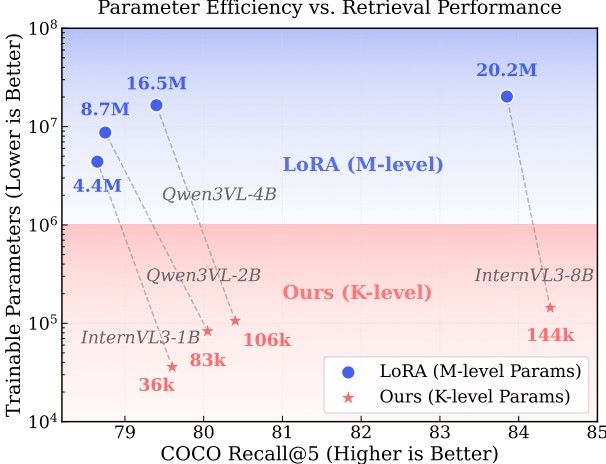

*Figure 1.* **Parameter Efficiency vs. Retrieval Performance.** We compare the number of trainable parameters against the retrieval performance, calculated as the average Recall@5 of COCO image-to-text and text-to-image retrieval. Our method achieves superior accuracy while tuning only thousands of parameters. This demonstrates an orders-of-magnitude efficiency gain compared to LoRA, which requires updating millions of parameters.

have recently shown strong multimodal understanding and reasoning abilities. Unlike traditional dual-tower retrieval models (Radford et al., 2021; Zhai et al., 2023; Li et al., 2022), which rely on separate encoders and are largely limited to coarse visual-text alignment, MLLMs employ a unified transformer that natively processes interleaved multimodal inputs and captures richer cross-modal semantic interactions. This advantage has motivated growing interest in adapting MLLMs for retrieval, with the goal of leveraging their powerful pre-trained representations.

Recognizing this potential, recent works (Li et al., 2026; Zhang et al., 2024b; Lin et al., 2024; Zhou et al., 2025) have explored using MLLMs as multimodal retrievers. The prevailing paradigm involves invasive strategies, such as full fine-tuning or LoRA (Hu et al., 2021), trained on massive multimodal datasets. However, this approach introduces significant challenges: (1) Semantic Degradation: Aligning generative MLLMs with discriminative contrastive objectives introduces an inherent objective mismatch, where

---

[1]School of Electronic Engineering, Beijing University of Posts and Telecommunications, China [2]Xiaohongshu Inc., China. Correspondence to: Chunxiao Fan <cxfan@bupt.edu.cn>.

| Image
Text | | | | | |
|---|---|---|---|---|---|
| A black cat. | Last token | 0.77 | 0.83 √ | 0.77 | 0.77 |
| | Query | 0.57 | 0.84 √ | 0.78 | 0.82 |
| An animal that barks | Last token | 0.72 | 0.79 | 0.81 √ | 0.71 |
| | Query | 0.66 | 0.77 | 0.84 √ | 0.76 |
| An animal has (2+7) lives | Last token | 0.80 × | 0.71 | 0.68 | 0.68 |
| | Query | 0.61 | 0.75 √ | 0.70 | 0.73 √ |

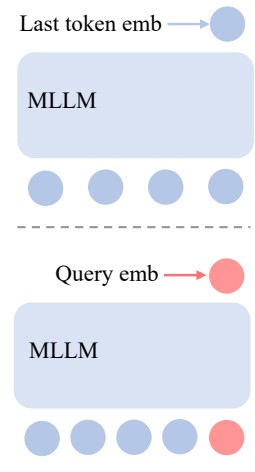

*(a)* Retrieval cases across three levels. Row 1: Pattern matching. Row 2: Knowledge Retrieval. Row 3: Logical Reasoning.

*(b)* Last token vs. Query

*Figure 2.* **Diagnostic pilot study.** We compare the zero-shot retrieval performance of the last token baseline against query token method on the InternVL3-1B backbone. The retrieval score based on cosine similarity is reported as the metric. The results suggest that the query token method better aggregates global context, enabling implicit reasoning for retrieval.

aggressive parameter updates can distort the pre-trained semantic space and induce catastrophic forgetting. (2) Training Inefficiency: Contrastive learning typically requires extremely large batch sizes to maintain negative sample diversity. Fine-tuning billion-parameter backbones under these conditions becomes a computationally prohibitive overhead. This raises a question: *Can an MLLM be transformed into a high-quality embedding model efficiently, without incurring the cost and semantic degradation of invasive fine-tuning?*

We posit that, through large-scale image–text pre-training, MLLMs (Bai et al., 2025a; Wang et al., 2025) have learned an aligned multimodal representation space. **From this perspective, retrieval adaptation is not about retraining the model but about eliciting its latent representations for retrieval**.

To validate this hypothesis, we designed a diagnostic pilot study using instruction prompts to enable zero-shot retrieval with a pre-trained MLLM. Specifically, we compared two aggregation strategies: (1) the last token baseline, which utilizes the final hidden states (e.g., <EOS>). (2) the query token method, in which a single zero-initialized query is appended to the input embedding sequence to aggregate contextual information, as shown in Figure 2b.

Figure 2a visualizes the retrieval performance across three increasing levels of difficulty: (Row 1) Pattern Matching: Both methods succeed in explicit matching. (Row 2) Knowledge Retrieval: For basic knowledge associations, the last token baseline suffers from low discriminability with indistinguishable confidence scores, whereas the query method maintains a high separation margin. (Row 3) Logical Reasoning: For complex logic such as "The animal with $2 + 7$ lives" (implying a cat), the last token fails entirely, retriev-

ing an irrelevant image due to its inability to process the arithmetic-to-visual reasoning chain. In contrast, the query method successfully performs implicit reasoning, correctly retrieving two distinct cat images.

These empirical results indicate that MLLMs inherently possess the ability to leverage internal knowledge and logical reasoning for retrieval. The last token baseline, however, suffers from an information bottleneck, as it struggles to compress complex semantic information into a single static representation. In contrast, a zero-initialized query yields meaningful, aligned semantic representations. The query (Pan et al., 2025) acts as a flexible aggregation mechanism, capturing global contextual information and producing more expressive representations for complex retrieval tasks.

Motivated by this insight, we propose **Shared Latent Queries (SLQ)**, a lightweight framework that empowers MLLMs into retrievers. Specifically, we introduce a set of learnable latent queries that are shared across image and text modalities, projecting both modalities into a unified embedding space. During training, we optimize only these learnable latent queries while freezing the backbone. Through the model's native causal attention mechanism, these learnable latent queries effectively elicit the MLLM's reasoning capabilities into compact embeddings. To rigorously validate these capabilities beyond superficial pattern matching, we construct the **Knowledge-Aware Reasoning Retrieval Benchmark (KARR-Bench)**, a diagnostic benchmark specifically designed to assess knowledge-aware reasoning capabilities.

To summarize, our main contributions are as follows:

- **Efficient MLLM-to-Retriever Adaptation:** We pro-

pose SLQ, an efficient framework that adapts frozen MLLMs for retrieval via Shared Latent Queries, while preserving pre-trained representations to leverage knowledge and reasoning for retrieval tasks.

- **Knowledge-Aware Reasoning Retrieval Benchmark:** We introduce KARR-Bench, a retrieval benchmark specifically designed to evaluate models' capabilities in knowledge-aware reasoning retrieval tasks.

- **Strong Performance with Minimal Overhead:** SLQ achieves superior multimodal retrieval performance with minimal parameter overhead, as illustrated in Figure 1.

## 2. Related Work

### 2.1. Vision-Language Representation Learning

Cross-modal retrieval relies on aligning visual and textual representations in a shared semantic space. Dual-tower architectures like CLIP (Radford et al., 2021) and ALIGN (Jia et al., 2021) employ contrastive learning on billion-scale image-text pairs, with subsequent works like BLIP (Li et al., 2022; 2023) and SigLIP (Zhai et al., 2023) improving data efficiency and training strategies. However, these methods encode modalities via disjoint encoders, limiting their ability to model deep cross-modal interactions and struggling with queries requiring complex reasoning or world knowledge.

### 2.2. Multimodal Large Language Models

MLLMs extend LLM reasoning capabilities to vision by mapping image features into the language model's embedding space (Liu et al., 2024a; Zhu et al., 2023; Dai et al., 2023). Recent large-scale models like GPT-4V (Achiam et al., 2023), Gemini (Team et al., 2024), Qwen-VL (Bai et al., 2023; Wang et al., 2024; Bai et al., 2025b;a), and InternVL (Chen et al., 2024; Zhu et al., 2025; Wang et al., 2025) demonstrate exceptional multimodal understanding through unified transformer architectures and massive pretraining. However, they are primarily optimized for autoregressive text generation rather than discriminative retrieval. Extracting high-quality, compact embeddings from these generative backbones without compromising their reasoning abilities remains an open challenge.

### 2.3. Adapting MLLMs for Multimodal Retrieval

Recent works repurpose MLLMs as dense retrievers. Methods like GME (Zhang et al., 2024b), MM-Embed (Lin et al., 2024), VLM2VEC (Jiang et al., 2024b), and MMRet (Zhou et al., 2025) extract the last token's hidden state, while VisRAG (Yu et al., 2024) and ColPali (Faysse et al., 2024) use multi-vector representations. These approaches typically employ full fine-tuning or LoRA, requiring massive com-

putation and risking semantic distortion. In contrast, SLQ utilizes learnable queries to aggregate features, avoiding invasive tuning while minimizing optimization cost.

### 2.4. Multimodal Prompt Tuning and Query-based Methods

Prompt tuning methods such as CoOp (Zhou et al., 2022), MaPLe (Khattak et al., 2023) and VPT (Jia et al., 2022) typically prepend learnable tokens to the input sequence. In bidirectional architectures, these tokens mainly serve as conditioning signals, while the final representation is taken from a summary token such as [CLS] for classification tasks. In contrast, SLQ appends learnable queries to the sequence end. Under causal attention in decoder-only MLLMs, these queries can attend to all preceding tokens, effectively acting as global embedding aggregators for retrieval.

Query-based methods such as BLIP-2 (Li et al., 2023) introduce additional modules (e.g., Q-Former) with crossattention for multimodal fusion. By contrast, SLQ performs aggregation entirely within the frozen MLLM using its native causal attention, without adding external modules.

More broadly, multimodal prompt learning primarily focuses on optimizing prompt conditions to adapt discriminative model like CLIP (Radford et al., 2021) to downstream domains. In contrast, SLQ is the first to demonstrate that frozen MLLMs can be transformed into powerful retrievers through a lightweight query-based interface, bridging the gap between generative pre-training and discriminative retrieval.

## 3. The KARR-Bench Benchmark

Current multimodal retrieval benchmarks largely rely on descriptive captions that map directly to visual features (e.g., matching "a red car" caption to an image that includes a red car). However, human-like intelligence involves the recognition of objects through implicit knowledge, logical inference, and cultural associations. To rigorously evaluate these capabilities, we construct the Knowledge-Aware Reasoning Retrieval Benchmark (KARR-Bench).

### 3.1. Construction Pipeline

We curate KARR-Bench from 5,000 images in the COCO test set via a three-stage pipeline: (1) Visual-grounded entity filtering removes abstract concepts to ensure all targets are visually verifiable. (2) Knowledge-enhanced query generation uses GPT-5-mini to encode target identities into implicit reasoning queries without explicit names or synonyms (Figure 3a), producing 4,500 candidate samples. (3) Human verification involves four annotators performing cross-validation to remove MLLM hallucinations and weakly cases. With a 60-70% acceptance rate, this step

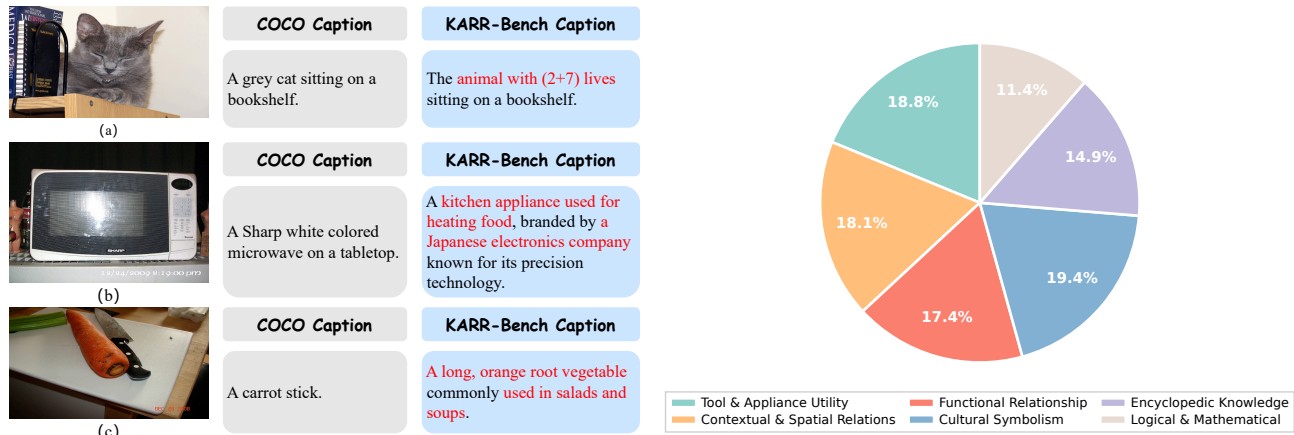

*(a)* Comparison between COCO and KARR-Bench      *(b)* Distribution of KARR-Bench Types

*Figure 3.* **Overview of KARR-Bench.** (a) Comparison between standard explicit captions and our knowledge reasoning captions. (b) The comprehensive distribution of categories in KARR-Bench.

ensures all queries are strictly grounded in visual evidence.

### 3.2. Dataset Statistics and Diversity

After filtering, KARR-Bench comprises 2,915 high-quality image-text pairs. Unlike benchmarks that rely on surface-level matching, KARR-Bench is designed to evaluate human-like reasoning and inference capabilities. As illustrated in Figure 3b, the queries span six dimensions: Tool & Appliance Utility (18.8%), Contextual & Spatial Relations (18.1%), Functional Relationship (17.4%), Cultural Symbolism (19.4%), Encyclopedic Knowledge (14.9%), and Logical & Mathematical (11.4%). This balanced distribution reduces domain bias and discourages shortcut exploitation. Detailed statistics are provided in Appendix E.

## 4. Method

### 4.1. Architecture Overview

As illustrated in Figure 4, SLQ consists of a frozen MLLM backbone and a small set of Shared Latent Queries. During training, all parameters of the MLLM are kept frozen to preserve its understanding and reasoning capabilities. For each image or text input, the Shared Latent Queries are appended to the end of the input sequence and jointly encoded via the backbone's causal attention mechanism. We extract the hidden states corresponding to these queries from the final transformer layer and apply mean pooling to obtain a compact embedding. This embedding is optimized via the contrastive learning objective to align vision and language representations in a unified space. At inference, these query embeddings serve as modality-agnostic representations for image-text retrieval.

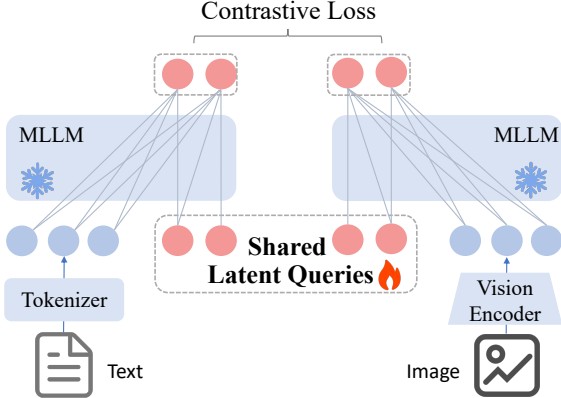

*Figure 4.* **Overview of the SLQ framework.** SLQ bridges the modality gap using a set of Shared Latent Queries that interact with the frozen MLLM via causal attention. The term "Shared" indicates that the same set of queries is appended to both image and text tokens, projecting them into a unified embedding space. During training, the MLLM backbone remains frozen, and only the queries are optimized via the contrastive objective to align vision and language representations.

### 4.2. Shared Latent Queries for Multimodal Retrieval

**Joint Encoding.** Let $T$ and $I$ denote a raw text and image input, respectively. The text embedding $\mathbf{E}_T$ is obtained from the MLLM's embedding layer, while the image embedding $\mathbf{E}_I$ is extracted from the vision encoder and projector.

The final input sequences are constructed by concatenating the multimodal embeddings $\mathbf{E}$, the instruction prompt embeddings $\mathbf{E}_P$, and the Shared Latent Queries $\mathbf{Q}$:

$$\mathbf{X}_T = [\mathbf{E}_T; \mathbf{E}_{P_T}; \mathbf{Q}], \tag{1}$$
$$\mathbf{X}_I = [\mathbf{E}_I; \mathbf{E}_{P_I}; \mathbf{Q}], \tag{2}$$

where $\mathbf{Q} \in \mathbb{R}^{N \times D}$, $N$ denotes the number of learnable queries, $D$ denotes the embedding dimension of the MLLM, and $[\cdot ; \cdot]$ denotes concatenation along the sequence dimension. Due to the backbone's causal attention mechanism, appending queries to the end allows them to attend to all preceding tokens, thereby facilitating global information aggregation for retrieval.

**Embedding Extraction.** The constructed sequences are processed by the frozen MLLM backbone $\mathcal{M}$ to obtain the final-layer hidden states:

$$\mathbf{H}_T = \mathcal{M}(\mathbf{X}_T), \quad \mathbf{H}_I = \mathcal{M}(\mathbf{X}_I). \quad (3)$$

Since the Shared Latent Queries $\mathbf{Q}$ are appended to the end of the sequence, we extract the hidden states corresponding to these queries, which correspond to the last $N$ positions:

$$\mathbf{H}_T^Q = \mathbf{H}_T[-N:], \quad \mathbf{H}_I^Q = \mathbf{H}_I[-N:]. \quad (4)$$

To obtain the final global representation for retrieval, we aggregate the information captured by the queries via mean pooling, followed by $\ell_2$-normalization:

$$\bar{\mathbf{h}}_T = \frac{1}{N} \sum_{k=1}^{N} \mathbf{h}_{T,k}^Q, \quad \mathbf{z}_T = \frac{\bar{\mathbf{h}}_T}{\|\bar{\mathbf{h}}_T\|_2}, \quad (5)$$

$$\bar{\mathbf{h}}_I = \frac{1}{N} \sum_{k=1}^{N} \mathbf{h}_{I,k}^Q, \quad \mathbf{z}_I = \frac{\bar{\mathbf{h}}_I}{\|\bar{\mathbf{h}}_I\|_2}, \quad (6)$$

where $\mathbf{h}_{T,k}^Q \in \mathbb{R}^D$ denotes the $k$-th feature vector within the query sequence (i.e., the $k$-th row of $\mathbf{H}_T^Q$). Finally, $\bar{\mathbf{h}}$ represents the aggregated query representation, and $\mathbf{z} \in \mathbb{R}^D$ is the final normalized embedding used for contrastive learning.

**Training Objective.** We optimize the Shared Latent Queries using a symmetric InfoNCE loss. Given a batch of $B$ paired image-text samples, we define the image-to-text and text-to-image losses as:

$$\mathcal{L}_{I2T} = -\frac{1}{B} \sum_{i=1}^{B} \log \frac{\exp(\langle \mathbf{z}_{I,i}, \mathbf{z}_{T,i} \rangle / \tau)}{\sum_{j=1}^{B} \exp(\langle \mathbf{z}_{I,i}, \mathbf{z}_{T,j} \rangle / \tau)}, \quad (7)$$

$$\mathcal{L}_{T2I} = -\frac{1}{B} \sum_{i=1}^{B} \log \frac{\exp(\langle \mathbf{z}_{T,i}, \mathbf{z}_{I,i} \rangle / \tau)}{\sum_{j=1}^{B} \exp(\langle \mathbf{z}_{T,i}, \mathbf{z}_{I,j} \rangle / \tau)}, \quad (8)$$

where $\langle \cdot, \cdot \rangle$ denotes the cosine similarity, and $\tau$ is the temperature parameter. The final training objective is the average of the bi-directional losses:

$$\mathcal{L} = \frac{1}{2}(\mathcal{L}_{I2T} + \mathcal{L}_{T2I}). \quad (9)$$

## 5. Experiments

**Evaluation Benchmarks.** We evaluate our method across a diverse set of benchmarks. For standard image-text retrieval, we adopt Flickr30K (Plummer et al., 2015) and COCO (Lin et al., 2014). To assess general multimodal embedding capability, we further evaluate on MMEB (Jiang et al., 2024b), a large-scale benchmark comprising 36 datasets that span four meta-tasks: classification, visual question answering (VQA), multimodal retrieval, and visual grounding, covering both in-distribution (IND) and out-of-distribution (OOD) settings. Finally, we benchmark our proposed KARR-Bench to measure knowledge-aware reasoning ability.

**Implementation Details.** We use InternVL3 (1B, 8B) (Zhu et al., 2025) and Qwen3-VL (2B, 4B) (Bai et al., 2025a) as backbone. To preserve pre-trained capabilities, the entire backbone is frozen during training, and only $N = 20$ Shared Latent Queries and the temperature parameter $\tau$ are optimized. For Flickr30K, COCO, and KARR-Bench, models are trained on the COCO training split for 5 epochs. For MMEB, we follow (Jiang et al., 2024b) and train on the MMEB-train for 1 epoch. We use a global batch size of 1024 for the 1B, 2B, and 4B models, and 512 for the 8B model. Comprehensive hyperparameter settings and training configurations are provided in Appendix A.

### 5.1. Main Results

**Performance on standard Retrieval.** Table 1 shows that SLQ achieves competitive performance on Flickr30K and COCO, validating the effectiveness of adapting frozen MLLMs for retrieval. Specifically, SLQ consistently outperforms dual-encoder retrievers in most metrics, with SLQ-1B even surpassing FLAME, which uses a 12B LLM for text encoding. Compared with MLLM-based baselines, SLQ demonstrates clear advantages. On COCO text-to-image retrieval, SLQ-1B remains competitive with VLM2VEC, whereas SLQ-8B significantly outperforms it, despite being trained on a smaller dataset.

**Performance on the MMEB.** Beyond standard retrieval, we evaluate SLQ on the diverse MMEB benchmark to assess its versatility across various multi-modal retrieval tasks, as shown in Table 2:

(1) Competitive performance and strong scalability. SLQ-8B achieves an Overall score of 67.5, outperforming VLM2VEC-7B (62.9), UniME-7B (66.6), MMRet-7B (64.1) and IDMR-8B (64.9). Notably, SLQ-4B (64.5) already surpasses VLM2VEC-7B and MMRet-7B with fewer parameters. In addition, performance improves steadily from 1B to 8B, demonstrating strong scalability of SLQ.

(2) Strong performance on VQA. SLQ performs well across

*Table 1.* **Performance comparison on Flickr30K and COCO retrieval benchmarks.** The best results are highlighted in **bold** and the second-best are underlined.

| Model | Flickr30K (1K test set) | | | | | | COCO (5K test set) | | | | | |
| | Image → Text | | | Text → Image | | | Image → Text | | | Text → Image | | |
| | R@1 | R@5 | R@10 | R@1 | R@5 | R@10 | R@1 | R@5 | R@10 | R@1 | R@5 | R@10 |
|---|---|---|---|---|---|---|---|---|---|---|---|---|
| *Dual-Encoder* | | | | | | | | | | | | |
| CLIP ViT-B (Radford et al., 2021) | 77.8 | 95.0 | 98.2 | 58.8 | 83.3 | 89.8 | 51.0 | 74.9 | 83.5 | 30.5 | 56.0 | 66.8 |
| CLIP ViT-L (Radford et al., 2021) | 87.2 | 98.3 | 99.4 | 67.3 | 89.0 | 93.3 | 58.1 | 81.0 | 87.8 | 37.0 | 61.6 | 71.5 |
| BLIP ViT-L (Li et al., 2022) | 75.5 | 95.1 | 97.7 | 70.0 | 91.2 | 95.2 | 63.5 | 86.5 | 92.5 | 48.4 | 74.4 | 83.2 |
| FLAME (Cao et al., 2025) | 86.4 | 97.3 | 98.6 | 73.3 | 91.7 | 95.5 | 60.5 | 82.9 | 89.3 | 43.9 | 70.4 | 79.7 |
| *MLLM-based* | | | | | | | | | | | | |
| E5-V-7B (Jiang et al., 2024a) | 88.2 | 98.7 | 99.4 | 79.5 | 95.0 | **97.6** | 62.0 | 83.6 | 89.7 | 52.0 | 76.5 | 84.7 |
| TIGeR (Qu et al., 2024) | - | - | - | 71.7 | 91.8 | 95.4 | - | - | - | 46.1 | 69.0 | 76.1 |
| VLM2VEC-7B (Jiang et al., 2024b) | **94.6** | **99.5** | **99.8** | 80.3 | 95.0 | 97.4 | 68.5 | 88.4 | 93.4 | 49.2 | 73.8 | 83.3 |
| *Ours* | | | | | | | | | | | | |
| SLQ (InternVL3-1B) | 86.7 | 97.8 | 99.6 | 74.4 | 92.9 | 95.9 | 61.2 | 84.9 | 91.8 | 48.5 | 74.3 | 83.1 |
| SLQ (Qwen3VL-2B) | 85.8 | 97.7 | 99.1 | 76.4 | 93.5 | 96.3 | 62.7 | 84.4 | 90.7 | 50.2 | 75.7 | 83.9 |
| SLQ (Qwen3VL-4B) | 85.9 | 97.9 | 99.6 | 76.8 | 93.4 | 96.1 | 64.3 | 85.6 | 91.2 | 50.4 | 75.2 | 83.4 |
| SLQ (InternVL3-8B) | 92.0 | 99.4 | **99.8** | **81.8** | **95.1** | **97.6** | **69.6** | **89.1** | **93.8** | **55.4** | **79.7** | **86.8** |

*Table 2.* **Results on the MMEB retrieval benchmarks.** Results are reported under two settings: * indicates fine-tuning on COCO, and † indicates fine-tuning on MMEB-train (Jiang et al., 2024b).

| Model | Classification (10) | VQA (10) | Retrieval (12) | Grounding (4) | IND (20) | OOD (16) | Overall (36) |
|---|---|---|---|---|---|---|---|
| BLIP2 (Li et al., 2023) | 27.0 | 4.2 | 33.9 | 47.0 | 25.3 | 25.1 | 25.2 |
| SigLIP (Zhai et al., 2023) | 40.3 | 8.4 | 31.6 | 59.5 | 32.3 | 38.0 | 34.8 |
| E5-V-7B (Jiang et al., 2024a) | 21.8 | 4.9 | 11.5 | 19.0 | 14.9 | 11.5 | 13.3 |
| MagiLens (Zhang et al., 2024a) | 38.8 | 8.3 | 35.4 | 26.0 | 33.1 | 23.7 | 27.8 |
| GME-7B (Zhang et al., 2024b) | 57.6 | 34.6 | **71.2** | 59.5 | - | - | 55.9 |
| MM-Embed-8B (Lin et al., 2024) | 48.1 | 32.3 | 63.8 | 57.8 | - | - | 50.0 |
| VLM2VEC-7B† (Jiang et al., 2024b) | **61.2** | 49.9 | 67.4 | 86.1 | 67.5 | 57.1 | 62.9 |
| UniME-7B† (Gu et al., 2025) | 60.6 | 52.9 | 67.9 | 85.1 | 68.4 | 57.9 | 66.6 |
| MMRet-7B† (Zhou et al., 2025) | 56.0 | 57.4 | 69.9 | 83.6 | 68.0 | 59.1 | 64.1 |
| IDMR-8B† (Liu et al., 2025) | 58.3 | 58.6 | 68.7 | 85.6 | **70.5** | 57.9 | 64.9 |
| SLQ-1B* | 33.9 | 11.0 | 35.0 | 53.5 | 30.8 | 29.1 | 30.1 |
| SLQ-2B* | 33.6 | 10.7 | 39.2 | 55.4 | 32.0 | 30.6 | 31.4 |
| SLQ-4B* | 35.4 | 11.3 | 43.7 | 56.2 | 34.5 | 32.6 | 33.9 |
| SLQ-8B* | 41.1 | 10.5 | 47.6 | 59.7 | 37.8 | 35.5 | 36.8 |
| SLQ-1B† | 52.5 | 49.1 | 60.2 | 78.3 | 59.8 | 54.5 | 57.3 |
| SLQ-2B† | 54.2 | 53.3 | 62.1 | 79.5 | 62.3 | 56.0 | 59.6 |
| SLQ-4B† | 55.7 | 58.7 | 65.4 | 84.2 | 66.5 | 58.6 | 64.5 |
| SLQ-8B† | 60.9 | **61.2** | 70.5 | **86.6** | 69.4 | **61.7** | **67.5** |

diverse retrieval tasks. Most notably, on VQA, SLQ-8B obtains 61.2, outperforming IDMR-8B (58.6), MMRet-7B (57.4), UniME-7B (52.9), and VLM2VEC (49.9) by a clear margin. We hypothesize that this advantage stems from freezing the MLLM backbone. Unlike full fine-tuning or LoRA-based adaptation, SLQ preserves pre-trained semantic understanding and reasoning abilities, which are particularly critical for VQA.

## 5.2. Tuning Strategies: Efficiency vs. Capability Preservation

We compare three tuning paradigms: (1) Full Fine-Tuning (Full FT) for LLM, where the vision encoder is frozen and only the LLM parameters are updated. (2) LoRA with rank $r = 8$ following VLM2VEC (Jiang et al., 2024b). (3) SLQ, which freezes the entire MLLM and optimizes only the learnable queries. As shown in Table 3, we evaluate them on both retrieval and VQA tasks, including MMMU (Yue et al., 2024), RealWorldQA (Zhang et al., 2024c), and OCRBench (Liu et al., 2024b), to assess the preservation of pre-trained capabilities.

**Capability Preservation.** Both Full FT and LoRA degrade the MLLM's inherent VQA capabilities. We attribute this to a fundamental optimization conflict: updating LLM parameters for contrastive retrieval disrupts its pre-trained autoregressive generation abilities. By freezing the backbone,

*Table 3.* **Comparison of tuning strategies.** We report Recall@5 for retrieve and assess the retention of the MLLM's inherent vision-language understanding capability on MMMU, RealWorldQA, and OCRBench. Training cost is measured in GPU hours on COCO for 5 epochs using H800 GPUs.

| Backbone | Dim. | Method | Param. | GPU Hour | Flickr30K | | COCO | | MMMU | RealWQA | OCRBench |
|---|---|---|---|---|---|---|---|---|---|---|---|
| | | | | | IR | TR | IR | TR | | | |
| InternVL3-1B | 896 | Full FT | 0.6B | 18.0 | 90.8 | 94.8 | 72.4 | 84.2 | 40.7 | 54.3 | 758 |
| | | LoRA | 4.4M | 11.4 | 90.4 | 95.7 | 72.9 | 84.4 | 42.1 | 56.8 | 785 |
| | | **SLQ** | **36k** | **3.9** | **92.9** | **97.8** | **74.3** | **84.9** | **43.4** | **58.2** | **790** |
| Qwen3VL-2B | 2048 | Full FT | 1.7B | 42.5 | 91.7 | 97.0 | 73.2 | 82.1 | 51.7 | 59.4 | 824 |
| | | LoRA | 8.7M | 23.8 | 92.1 | 97.2 | 74.8 | 82.7 | 53.1 | 62.8 | 832 |
| | | **SLQ** | **83K** | **6.7** | **93.5** | **97.7** | **75.7** | **84.4** | **53.4** | **63.9** | **858** |
| Qwen3VL-4B | 2560 | Full FT | 4B | 96.2 | 92.0 | 97.2 | 74.5 | 82.8 | 63.8 | 66.7 | 851 |
| | | LoRA | 16.5M | 48.6 | 92.7 | 96.8 | 75.1 | 83.7 | 65.4 | 68.3 | 857 |
| | | **SLQ** | **106K** | **13.5** | **93.4** | **97.9** | **75.2** | **85.6** | **67.4** | **70.9** | **881** |
| InternVL3-8B | 3584 | Full FT | 7.6B | 403.8 | 94.0 | 96.6 | 78.2 | 87.4 | 59.8 | 66.5 | 847 |
| | | LoRA | 20.2M | 130.1 | 94.4 | 98.7 | 79.4 | 88.3 | 60.9 | 67.2 | 843 |
| | | **SLQ** | **144k** | **38.9** | **95.1** | **99.4** | **79.7** | **89.1** | **62.7** | **70.8** | **880** |

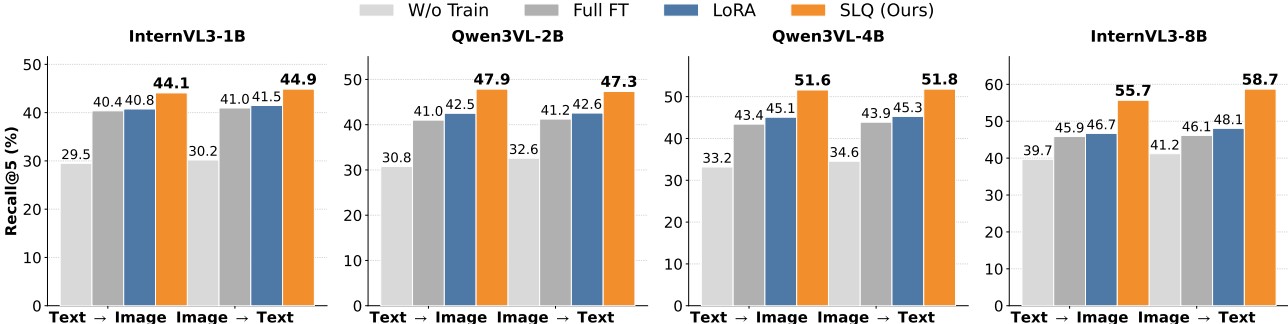

*Figure 5.* **Performance comparison on KARR-Bench.** Left to Right: Results on InternVL3-1B, Qwen3VL-2B, Qwen3VL-4B, and InternVL3-8B. Our method (shown in **Orange**) consistently outperforms both invasive tuning baselines. Specifically, on the strongest InternVL3-8B backbone, our method achieves significant gains over LoRA and Full FT. These results indicate that preserving a frozen backbone while SLQ is effective for knowledge-aware reasoning retrieval.

SLQ successfully circumvents this catastrophic forgetting.

**Retrieval & Efficiency.** SLQ achieves superior retrieval performance with drastically reduced training costs. Across all scales, SLQ consistently outperforms LoRA and Full FT. The fact that LoRA beats Full FT aligns with prior findings (Zhang et al., 2024b; Jiang et al., 2024b) that invasive parameter updates are suboptimal for adapting generative models to representation tasks. Crucially, SLQ maximizes efficiency with trainable parameter: for SLQ-1B, SLQ requires only 36K parameters and cuts training time by 66%. On SLQ-8B, SLQ uses just 144K parameters and reduces GPU hours by 70%. This demonstrates that SLQ offers a highly efficient, non-destructive adaptation path.

### 5.3. Knowledge-Aware Reasoning Retrieval

Figure 5 shows results on KARR-Bench. As model scale increases, a clear gap emerges: while Full FT and LoRA show diminishing gains, SLQ scales and benefits more from larger backbones.

The gap between SLQ and LoRA widens from 3.3% (1B) to 9.8% (8B), while both baselines yield only marginal gains over the untrained model at larger scales. This suggests that larger MLLMs encode richer yet more fragile world knowledge, which can be disrupted during adaptation. By keeping the backbone frozen, SLQ better preserves these capabilities, highlighting the importance of maintaining pretrained representations as model scale grows.

### 5.4. Analysis of Modality Gap and Alignment

Following standard practices in recent representation studies (Liang et al., 2022; Jiang et al., 2024a; Zhang et al., 2024b), we visualize embedding distributions via PCA and quantitatively evaluate them using three metrics: modality gap ($\|gap\|$, defined as the Euclidean distance between image and text centroids), Alignment (Wang & Isola, 2020), and Uniformity (Wang & Isola, 2020). Lower Alignment

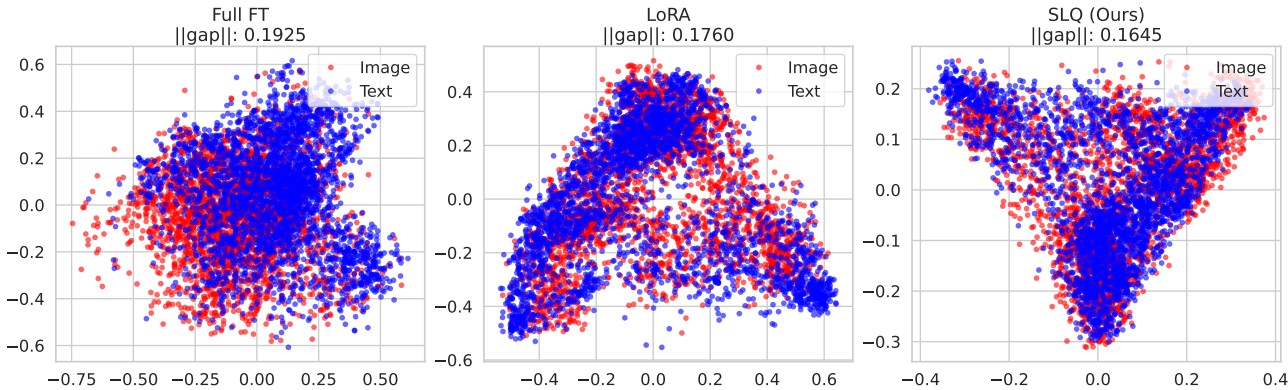

*Figure 6.* **Visualization of the unified representation space using PCA.** Red points represent Image embeddings, and Blue points represent Text embeddings. Full FT and LoRA exhibit a noticeably broader spatial spread. In contrast, SLQ maintains a much more *compact distribution*, resulting in a smaller centroid distance gap and demonstrating superior cross-modal alignment.

*Table 4.* **Analysis of representation geometry.** Lower (↓) Alignment and ‖gap‖ indicate better matching. Lower Uniformity indicates evenly distributed spaces.

| Method | ‖gap‖ (↓) | Alignment (↓) | Uniformity (↓) | |
| --- | --- | --- | --- | --- |
| | | | Text | Image |
| Full FT | 0.193 | 0.182 | -1.15 | -1.09 |
| LoRA | 0.176 | 0.155 | -1.30 | -1.26 |
| **SLQ (Ours)** | **0.165** | **0.148** | **-1.45** | **-1.42** |

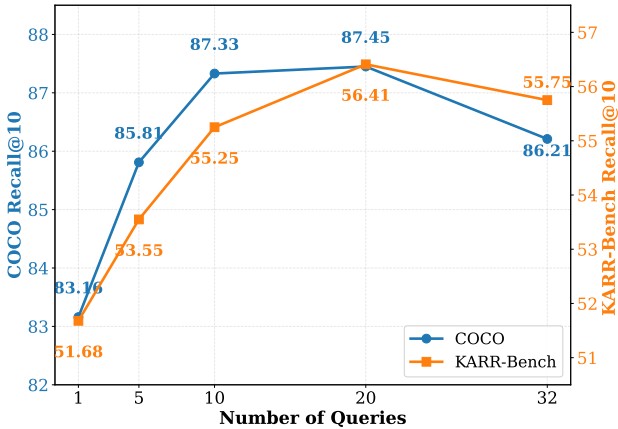

*Figure 7.* **Ablation on the number of queries.** The left axis shows the performance on COCO while the right axis shows on KARR-Bench.

indicates better positive-pair matching, while lower Uniformity indicates more evenly distributed representations on the hypersphere.

As shown in Figure 6, compared to the broader spatial separation between modalities seen in Full FT and LoRA, SLQ exhibits a more compact alignment between image and text embeddings. This observation aligns well with the quantitative results in Table 4, where SLQ achieves the smallest modality gap and alignment error, while simultaneously maintaining the lowest uniformity across both modalities.

### 5.5. Ablation Studies

**Impact of the Number of Queries.** We investigate the impact of the query count $N \in \{1, 5, 10, 20, 32\}$ in Figure 7. Performance follows a distinct trajectory that rises before declining, consistently peaking at $N = 20$. Increasing $N$ to 32 leads to degradation, suggesting that excessive queries introduce redundancy and overfitting. Crucially, the growth trends differ across datasets. While COCO saturates early, reasoning-heavy tasks like KARR-Bench show continuous improvement up to $N = 20$. We attribute this to the queries serving as a higher-capacity aggregator that captures multi-step semantic dependencies, thereby improving complex retrieval.

**Comparison with PEFT.** In Table 5, we first compare SLQ with Parameter-Efficient Fine-Tuning (PEFT) methods. Among the evaluated tuning paradigms, Prompt Tuning (Khattak et al., 2023) performs the worst, likely due to causal masking, which restricts interaction between learned query and input sequence. Adapter Tuning (Sung et al., 2022) underperforms LoRA, suggesting limited effectiveness in interacting with pre-trained MLLM representations. In contrast, SLQ consistently outperforms all PEFT baselines, demonstrating that carefully designed queries can effectively extract multimodal representations.

**Ablation on Extraction Strategies** We investigate two categories of extraction designs in Table 5: (1) External Projection Heads, which apply a linear layer or a Transformer block (TF Block) to the last token hidden states. (2) Separate Queries, which assign independent learnable queries to visual and textual inputs respectively.

*Table 5.* All ablations are conducted using InternVL3-1B and report Recall@5 on COCO.

| Category | Design | I → T | T → I |
|---|---|---|---|
| PEFT Strategy | LoRA | 84.4 | 72.9 |
| | Prompt Tuning (Khattak et al., 2023) | 82.2 | 69.7 |
| | Adapter Tuning (Sung et al., 2022) | 83.2 | 71.0 |
| Extraction Design | Head (Linear) | 78.7 | 67.4 |
| | Head (TF Block) | 78.2 | 66.7 |
| | Separate Queries | 83.2 | 72.5 |
| Pooling Strategy | Last Token | 84.4 | 74.1 |
| | Max Pooling | 84.1 | 73.8 |
| | **SLQ(Mean Pooling)** | **84.9** | **74.3** |

Both head-based designs perform poorly, and the more complex TF Block does not improve over the linear. This supports the information bottleneck hypothesis and suggests that external heads, operating outside the MLLM, cannot effectively leverage its internal multimodal representations. "Separate Queries" performs better than these heads but still underperforms SLQ, despite using more parameters. We attribute this to the benefit of parameter sharing: shared queries encourage both modalities to align in a unified embedding space, while separate queries tend to produce weaker cross-modal alignment. Overall, these results highlight that SLQ offers a simple yet effective extraction strategy.

**Impact of Pooling Strategies.** We compare Last, Max, and Mean pooling for aggregating the $N$ learned queries. As shown in the bottom section of Table 5, Mean Pooling performs best, likely because it balances information across all query positions without being dominated by any single representation.

### 5.6. Additional Evaluations and Qualitative Analysis.

A comparison between VLM2VEC and SLQ on KARR-Bench is presented in Appendix B. Detailed LoRA configurations and corresponding ablation studies are provided in Appendix C. We further extend our experiments to composed and image-to-image retrieval, with detailed results provided in Appendix D. A qualitative case study on KARR-Bench is presented in Appendix G.

## 6. Limitations

A core design of SLQ is to keep the MLLM backbone frozen, preserving pre-trained capabilities and avoiding catastrophic forgetting. However, this design may limit adaptation to extreme out-of-distribution domains, such as specialized medical or satellite imagery, where learning new visual features is required. In such cases, partial unfreezing or combining SLQ with lightweight visual adapters may be beneficial. Moreover, SLQ is currently validated only under the symmetric InfoNCE objective, leaving its effectiveness under

alternative training paradigms—such as triplet loss, listwise ranking, or knowledge distillation—unexplored. Extending SLQ to broader architectures, incorporating harder negative sampling strategies, and exploring diverse objectives are promising directions for future work.

## 7. Conclusion

In this paper, we introduce SLQ, a parameter-efficient framework that unlocks the retrieval potential of MLLMs while preserving their pre-trained knowledge and reasoning capabilities. By freezing the backbone and learning only a lightweight set of shared latent queries, SLQ effectively aligns modalities and maintains the pre-trained semantic space. Extensive experiments on COCO, Flickr, MMEB and our proposed KARR-Bench demonstrate that SLQ not only reduces trainable parameters by orders of magnitude but also reduces training cost, while achieving strong performance compared to baselines. These results highlight the importance of preserving pre-trained representations and establish SLQ as an effective and efficient approach for MLLM-based retrieval.

## Acknowledgement

This work is supported in part by the National Natural Science Foundation of China (Grant Nos. 62376034 and 92467105), Beijing Natural Science Foundation(Grant No. L241011) and State Grid Corporation of China Headquarters Project (Grant No.5700-202458331A-2-1-ZX).

## Impact Statement

This paper presents work whose goal is to advance the field of Machine Learning. There are many potential societal consequences of our work, none which we feel must be specifically highlighted here.

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

## Appendix Overview

This supplementary document is organized as follows:

- Sec. A shows the experimental settings.

- Sec. B compares the performance of VLM2VEC and SLQ on KARR-Bench.

- Sec. C provides the detailed LoRA configurations and corresponding ablation studies.

- Sec. D shows the experiment on image-image retrieval and composed image retrieval.

- Sec. E shows the KARR-Bench detailed statistics.

- Sec. F shows the prompts for KARR-Bench construction.

- Sec. G shows the qualitative case study on KARR-Bench.

## A. Experimental Settings

For global optimization, we consistently employ the AdamW optimizer coupled with a cosine decay learning rate schedule. A warmup ratio of 0.03 is applied to stabilize the early training phase. To enhance memory efficiency and training speed, we utilize DeepSpeed with the ZeRO-2 stage optimization. For experiments involving LoRA, the rank is set to 8. Specifically, InternVL3 is configured with a fixed input resolution of 448×448. In contrast, Qwen-3-VL supports dynamic resolutions, allowing it to process images in their native aspect ratios without any distortion. The specific configurations are detailed in Tables 6 and 7.

*Table 6.* Global optimization hyperparameters. These settings were applied consistently across all experiments to ensure a fair comparison.

| Hyperparameter | Value |
|---|---|
| Optimizer | AdamW |
| LR Schedule | Cosine Decay |
| Warmup Ratio | 0.03 |
| DeepSpeed | ZeRO-2 |
| LoRA Rank | 8 |
| LoRA Alpha | 16 |
| LoRA Dropout | 0.05 |

As shown in Table 7, for the 1B model, SLQ drastically reduces the number of trainable parameters (only 36K) and requires merely 3.9 GPU hours, saving roughly 65% of training time compared to LoRA (11.4 GPU hours). For the 8B model, SLQ achieves an even greater time saving of 70% compared to LoRA (38.9 vs. 130.1 GPU hours). Furthermore, SLQ significantly lowers the overall hardware requirements, allowing us to halve the GPU count needed for both 1B and 8B scales.

## B. SOTA Comparison on KARR-Bench

To further demonstrate the difficulty of our proposed KARR-Bench, we evaluate the recent state-of-the-art model VLM2Vec-Qwen2VL-7B (Jiang et al., 2024b). As shown in Table 8, VLM2Vec-7B achieves 49.9 and 50.1 on Image-to-Text and Text-to-Image Recall@5, respectively. However, it underperforms our SLQ-4B model. This result indicates that KARR-Bench remains highly challenging even for recent state-of-the-art models, highlighting the effectiveness of our proposed method in complex retrieval scenarios.

## C. LoRA Configurations and Ablation Study

**LoRA Configuration.** In our main experiments, we apply Low-Rank Adaptation (LoRA) to all linear layers of the LLM. Following GME (Zhang et al., 2024b), we set the LoRA rank $r = 8$, $\alpha = 16$, and the dropout rate to 0.05.

*Table 7.* Hyperparameter settings and training costs for different backbones.

| Backbone | Dim | Method | Bsz / GPU | Total GPUs | Global Bsz | LR | Train Time | GPU Hours |
|---|---|---|---|---|---|---|---|---|
| InternVL3-8B | 3584 | Full FT | 16 | 32×H800 | 512 | 1e-5 | 12:37:02 | 403.8 |
| | | LoRA | 16 | 32×H800 | 512 | 1e-5 | 4:03:49 | 130.1 |
| | | SLQ | 32 | 16×H800 | 512 | 5e-4 | 2:25:53 | 38.9 |
| Qwen3VL-4B | 2560 | Full FT | 32 | 32×H800 | 1024 | 1e-5 | 3:00:22 | 96.2 |
| | | LoRA | 32 | 32×H800 | 1024 | 1e-5 | 1:31:08 | 48.6 |
| | | SLQ | 64 | 16×H800 | 1024 | 5e-4 | 0:50:37 | 13.5 |
| Qwen3VL-2B | 2048 | Full FT | 32 | 32×H800 | 1024 | 1e-5 | 1:19:41 | 42.5 |
| | | LoRA | 32 | 32×H800 | 1024 | 1e-5 | 0:44:36 | 23.8 |
| | | SLQ | 64 | 16×H800 | 1024 | 5e-4 | 0:25:07 | 6.7 |
| InternVL3-1B | 896 | Full FT | 64 | 16×H800 | 1024 | 1e-5 | 1:07:38 | 18.0 |
| | | LoRA | 64 | 16×H800 | 1024 | 1e-5 | 0:42:57 | 11.4 |
| | | SLQ | 128 | 8×H800 | 1024 | 5e-4 | 0:29:32 | 3.9 |

*Table 8.* SOTA comparison on KARR-Bench (Recall@5).

| Method | Image → Text | Text → Image |
|---|---|---|
| VLM2VEC-7B (Jiang et al., 2024b) | 49.9 | 50.1 |
| SLQ-1B (Ours) | 44.9 | 44.1 |
| SLQ-2B (Ours) | 47.3 | 47.9 |
| SLQ-4B (Ours) | 51.8 | 51.6 |
| SLQ-8B (Ours) | **58.7** | **55.7** |

**Ablation on LoRA Ranks.** Additionally, we conduct ablation experiments to analyze the impact of the LoRA rank by varying $r \in \{4, 8, 16, 32\}$. For this ablation study, we use the InternVL-1B backbone and report the Recall@5 performance on the COCO dataset.

As shown in Table 9, we observe that higher LoRA ranks do not lead to performance improvements. Furthermore, full fine-tuning generally performs worse than LoRA, which is consistent with the findings in GME (Zhang et al., 2024b) and VLM2Vec (Jiang et al., 2024b). This phenomenon suggests that increasing the number of trainable parameters exacerbates the misalignment with pre-trained representations. This is likely due to the inherent conflict between the contrastive learning objectives used during fine-tuning and the generative pre-training objectives of the backbone.

In contrast, our proposed SLQ method freezes the MLLM backbone entirely, thereby successfully avoiding this misalignment issue and consistently outperforming the LoRA-based adaptations.

*Table 9.* Ablation on LoRA ranks on COCO (Recall@5), using InternVL-1B.

| Method | Rank | Image → Text | Text → Image |
|---|---|---|---|
| LoRA | 4 | 84.5 | 72.5 |
| LoRA | 8 | 84.4 | 72.9 |
| LoRA | 16 | 83.9 | 72.2 |
| LoRA | 32 | 82.1 | 71.6 |
| SLQ (Ours) | — | **84.9** | **74.3** |

*Table 10.* Zero-shot composed image retrieval performance on FashionIQ (Average R@50) and CIRR (R@5). * indicates E5-V trained with image–text pairs.

| Method | FashionIQ (Avg R@50) | CIRR (R@5) |
|---|---|---|
| E5-V-7B* (Jiang et al., 2024a) | 30.8 | 33.5 |
| InternVL3-1B (Full FT) | 32.7 | 53.2 |
| InternVL3-1B (LoRA) | 33.1 | 51.4 |
| InternVL3-1B (SLQ) | 35.0 | 57.8 |
| InternVL3-8B (Full FT) | 37.6 | 55.7 |
| InternVL3-8B (LoRA) | 39.3 | 56.8 |
| InternVL3-8B (SLQ) | 43.1 | 63.5 |

## D. Experiment on Image-Image Retrieval and Composed Image Retrieval

**Zero-shot Image-to-Image Retrieval.** Table 11 compares the zero-shot retrieval performance on the I2I-Flickr30K and I2I-COCO datasets. We evaluate performance across two distinct tasks: standard image retrieval and text (rendered as image) retrieval, reporting R@1, R@5, and R@10 metrics.

The results highlight the significant advantages of our SLQ framework: Superiority over Baselines: On the large-scale InternVL3-8B backbone, SLQ achieves substantial improvements over E5-V baseline. Specifically, SLQ achieves an R@1 of 80.9% on I2I-Flickr30K image retrieval, surpassing E5-V (67.8%) by a remarkable margin of 13.1%. Similarly, on I2I-COCO, SLQ outperforms E5-V by 14.2% (55.4% vs. 41.2%). Advantage over Traditional Fine-tuning: Compared to Full FT and LoRA, SLQ consistently delivers higher retrieval accuracy. For instance, on the 8B model, SLQ outperforms LoRA by 11.9% on I2I-Flickr30K (R@1). This indicates that our non-invasive approach effectively preserves the pre-trained knowledge of the backbone while adapting to retrieval tasks, whereas invasive tuning may suffer from catastrophic forgetting or suboptimal alignment in the zero-shot setting. Robustness in Typographic Understanding: In the "text (render as image)" retrieval task, which requires strong optical character recognition (OCR) and semantic understanding capabilities, SLQ demonstrates exceptional performance. On I2I-COCO, SLQ (8B) achieves 69.6% R@1, significantly outperforming both E5-V (51.6%) and Full FT (57.6%). This confirms that keeping the vision encoder frozen is crucial for maintaining the fine-grained visual perception capabilities required for typographic understanding.

**Zero-shot Composed Image Retrieval.** Table 10 presents the zero-shot performance on the FashionIQ (Wu et al., 2021) and CIRR (Liu et al., 2021) benchmarks. We compare our proposed SLQ method against the baseline E5-V and standard fine-tuning strategies (Full FT and LoRA) across different model scales (InternVL3-1B and 8B).

As shown in Table 10, SLQ consistently achieves superior performance compared to other methods. Specifically: On the InternVL3-1B backbone, SLQ outperforms Full FT by 2.3% on FashionIQ (R@50) and 4.6% on CIRR (R@5). The improvement is even more significant on the larger InternVL3-8B model, where SLQ surpasses Full FT by 5.5% on FashionIQ and 7.8% on CIRR, demonstrating the scalability and effectiveness of our approach. Furthermore, our method significantly outperforms E5-V, highlighting the advantage of the proposed SLQ mechanism in the zero-shot retrieval setting.

*Table 11.* Zero-shot image-image retrieval performance on I2I-Flickr30K and I2I-COCO.

| Method | image retrieval | | | | | | text (render as image) retrieval | | | | | |
|---|---|---|---|---|---|---|---|---|---|---|---|---|
| | I2I-Flickr30K | | | I2I-COCO | | | I2I-Flickr30K | | | I2I-COCO | | |
| | R@1 | R@5 | R@10 | R@1 | R@5 | R@10 | R@1 | R@5 | R@10 | R@1 | R@5 | R@10 |
| E5-V-7B (Jiang et al., 2024a) | 67.8 | 89.2 | 93.6 | 41.2 | 66.7 | 76.2 | 79.5 | 95.2 | 97.8 | 51.6 | 76.8 | 84.9 |
| InternVL3-1B (Full FT) | 61.2 | 84.3 | 89.7 | 39.1 | 65.4 | 75.8 | 73.5 | 91.3 | 96.0 | 49.7 | 77.3 | 84.2 |
| InternVL3-1B (LoRA) | 60.4 | 84.6 | 90.2 | 39.0 | 66.0 | 76.5 | 73.3 | 91.8 | 96.3 | 49.7 | 77.0 | 85.3 |
| InternVL3-1B (SLQ) | 64.8 | 87.3 | 92.2 | 41.2 | 67.7 | 77.9 | 76.9 | 95.0 | 97.2 | 54.1 | 80.0 | 87.7 |
| InternVL3-8B (Full FT) | 64.9 | 86.5 | 91.7 | 44.5 | 70.6 | 79.3 | 79.4 | 94.9 | 98.1 | 57.6 | 78.2 | 88.7 |
| InternVL3-8B (LoRA) | 69.0 | 88.7 | 93.8 | 46.7 | 73.3 | 82.7 | 83.1 | 96.2 | 97.9 | 61.0 | 83.0 | 89.8 |
| InternVL3-8B (SLQ) | 80.9 | 94.9 | 97.1 | 55.4 | 79.7 | 86.8 | 90.2 | 99.1 | 99.6 | 69.6 | 89.1 | 93.8 |

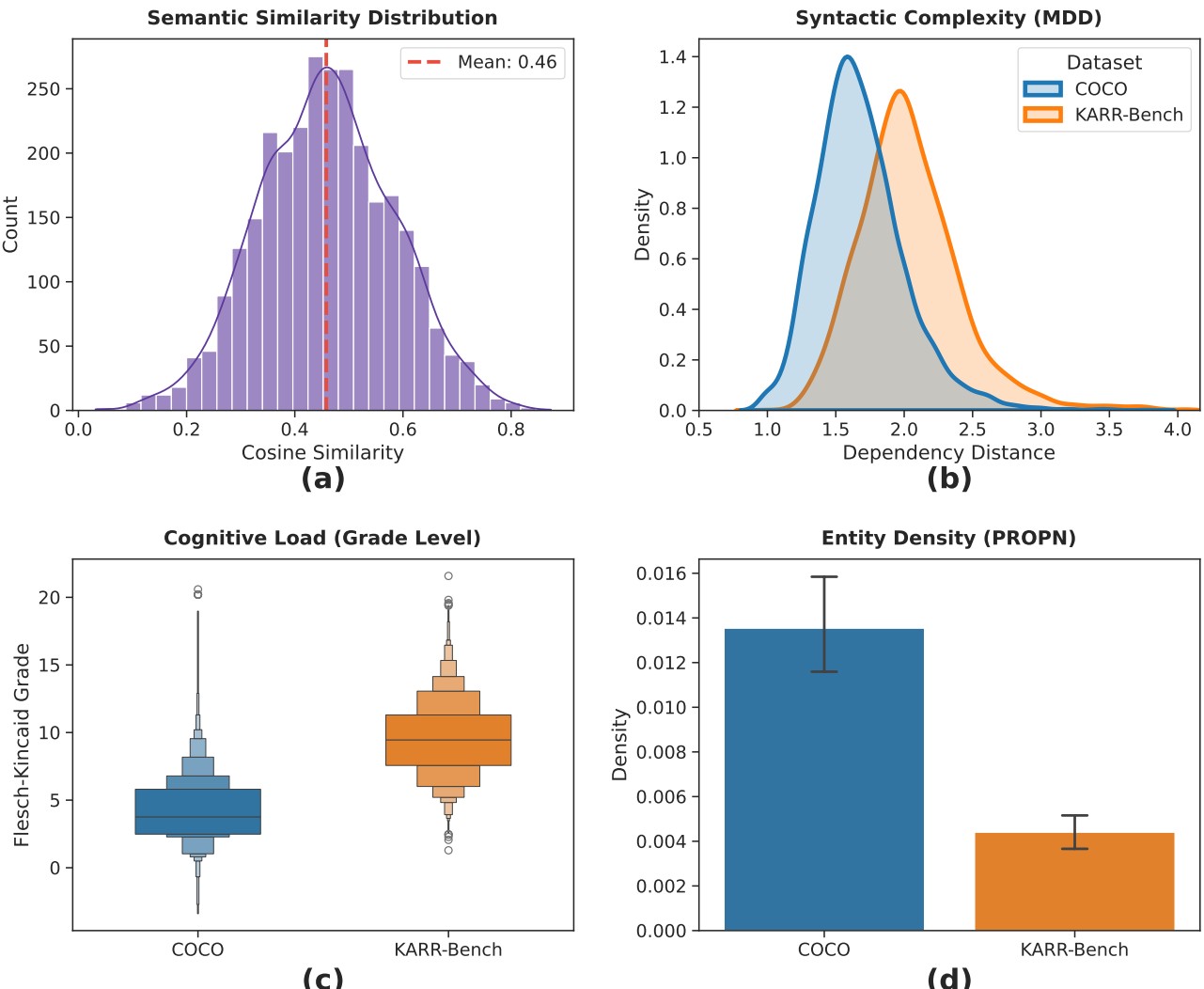

*Figure 8.* **Linguistic and Semantic Comparison between COCO and KARR-Bench.** (a) The cosine similarity distribution demonstrates that KARR-Bench Captions maintain semantic relevance to the visual content despite abstraction. (b) KARR-Bench exhibits significantly higher Syntactic Complexity (MDD). (c) The Cognitive Load (Flesch-Kincaid Grade Level) doubles from COCO to KARR-Bench, indicating a shift to advanced reading comprehension. (d) The drastic drop in Entity Density (Proper Nouns) confirms the removal of explicit naming cues.

## E. KARR-Bench Detailed Statistics

In this section, we provide a comprehensive quantitative analysis to verify that KARR-Bench effectively transitions from explicit visual descriptions to knowledge-aware reasoning. By comparing our benchmark against standard COCO captions, we observe distinct shifts in linguistic complexity, semantic abstraction, and reasoning diversity.

**Syntactic Complexity and Cognitive Load.** The transition from descriptive captioning to reasoning-based retrieval necessitates a fundamental increase in linguistic complexity. As illustrated in Figure 8(b), KARR-Bench Captions exhibit a notably higher Mean Dependency Distance (MDD) (Liu et al., 2017) compared to COCO. This metric suggests that our queries move beyond simple subject-verb-object structures, employing complex grammatical dependencies to encode logic. This structural sophistication directly translates to increased cognitive demand; Figure 8(c) reveals that the Flesch-Kincaid Grade Level (Solnyshkina et al., 2017) rises from approximately Grade 5 (simple English) in COCO to Grade 10 in KARR-Bench. Consequently, successfully parsing these queries requires models to possess advanced language understanding capabilities akin to high-school level reading comprehension.

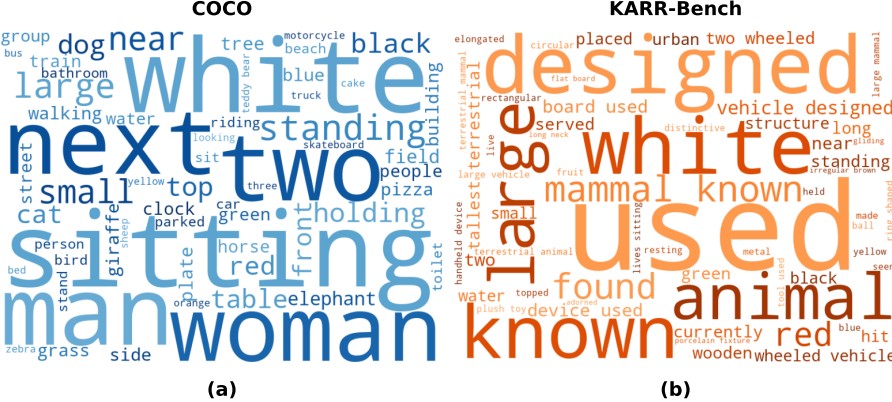

*Figure 9.* **Word Cloud Comparison.** The vocabulary shifts from observational primitives in COCO (a) to abstract, functional, and relational terms in KARR-Bench (b), reflecting the requirement for implicit reasoning.

**Entity Masking and Semantic Integrity.** A critical design objective of KARR-Bench is to prevent models from relying on trivial keyword matching or Named Entity Recognition (NER). We validate the efficacy of our filtering protocol through the density of Proper Nouns (PROPN), which shows a sharp decline in Figure 8(d). This confirms that explicit names (e.g., brand names, specific locations) have been successfully masked, forcing the model to identify objects based on their attributes and history. Despite this abstraction, the queries do not drift into hallucination. The cosine similarity distribution in Figure 8(a) shows a mean similarity of 0.46 with original captions, indicating that while the phrasing is distinct, the queries remain semantically grounded in the same visual reality as the source images.

**Lexical Shift: From Appearance to Function.** The qualitative difference between standard captions and reasoning queries is visually palpable in the vocabulary distributions shown in Figure 9. COCO captions are dominated by concrete visual primitives and observational nouns such as *"white", "sitting", "man",* and *"dog."* In contrast, KARR-Bench exhibits a marked lexical shift towards abstract, functional, and associative terminology. Prominent terms include *"designed", "used", "known",* and *"symbol,"* underscoring that the benchmark evaluates *why* an object exists or *what* it does, rather than merely describing its surface appearance. This shift verifies that KARR-Bench successfully decouples linguistic queries from explicit visual cues.

**Diversity of Reasoning Logic.** Finally, we analyze the composition of reasoning strategies required to solve the benchmark. As shown in Figure 3b, unlike traditional datasets that rely predominantly on simple visual pattern matching, KARR-Bench exhibits a highly balanced distribution across six distinct cognitive dimensions. Queries requiring physical and practical understanding form a solid foundation, comprising **Tool & Appliance Utility** (18.8%), **Contextual & Spatial Relations** (18.1%), and **Functional Relationship** (17.4%). These categories test a model's ability to infer how objects are used, their spatial positioning, and how multiple entities interact within a scene. Furthermore, the benchmark heavily evaluates the retrieval of external world knowledge not present in the pixel space: **Cultural Symbolism** (19.4%) and **Encyclopedic Knowledge** (14.9%) require models to bridge visual cues with abstract meanings, societal context, and factual knowledge. Additionally, **Logical & Mathematical** queries (11.4%) rigorously assess higher-order deductive reasoning and arithmetic capabilities. Crucially, this balanced distribution effectively reduces domain bias and discourages models from exploiting superficial dataset shortcuts. By ensuring that no single type of inference dominates, KARR-Bench provides a comprehensive and robust evaluation of human-like visual reasoning and intelligence.

## F. Prompts for KARR-Bench Construction

To ensure the reproducibility of our **KARR-Bench**, we provide the detailed prompts used in our construction pipeline. As described in Section 3, the process is divided into two distinct logical stages: (1) Visual-Grounded Entity Filtering, and (2) Knowledge-Enhanced Query Generation. We utilized "gpt-5-mini-2025-08-07" for both stages.

**Instruction:** Given an image caption, select *at most one* core entity (head noun) that can serve as a reliable target for knowledge-based visual reasoning.

The selected entity must satisfy all of the following: (i) it refers to a concrete object mentioned explicitly in the caption; (ii) it admits a stable commonsense or encyclopedic definition; (iii) it does not depend on unverifiable image-specific details such as identity, brand, or event context.

**Rejection Criteria:** Output `None` if the caption only contains entities from the following categories:

1. Generic human references (e.g., "man", "woman", "person", "people", "crowd").

2. Background or environmental elements (e.g., "wall", "floor", "sky", "grass", "street", "scene", "view").

3. Abstract concepts, actions, or events (e.g., "performance", "activity", "event", "situation").

4. Low-semantic or ambiguous objects (e.g., "object", "thing", "shape", "line").

**Constraint:** Do not infer or extrapolate object categories beyond what can be determined with certainty from the caption and image.
**Output:** If a valid entity exists, output its name. Otherwise, output `None`.

*Table 12.* Stage 1 prompt for KARR-Bench construction.

**Instruction:** Given a validated target entity, rewrite it into an implicit query whose resolution requires reasoning or external knowledge rather than direct visual matching.

The rewritten description should function as a riddle-style reference that allows the original entity to be inferred through knowledge or logic.

**Allowed Reasoning Types (choose at least one):**

- **Logical or numerical reasoning:** defining the entity via arithmetic or symbolic properties.

- **Functional commonsense:** describing the entity by its typical use or purpose.

- **Cultural or symbolic association:** referring to widely recognized cultural or historical meanings.

- **Encyclopedic definition:** using dictionary-style or factual descriptions grounded in stable knowledge.

- **Contextual & Spatial Relations:** describing entities based on their physical positioning, layout, or environmental context within a scene.

- **Functional Relationship:** defining the interactions, dependencies, or cause-and-effect relationships between multiple entities.

**Strict Constraints:**

1. Do not include the entity name or any of its morphological variants.

2. Do not use direct synonyms or trivial paraphrases.

3. Avoid describing visual appearance (e.g., color or shape) unless required by the reasoning itself.

**Rejection Rule:** If the entity cannot be described using stable and widely accepted knowledge, output `CANNOT_REWRITE`.

*Table 13.* Stage 2 prompt for KARR-Bench construction.

## F.1. Stage 1: Entity Extraction and Filtering

In the first stage, the model is instructed to extract the most salient head noun from the raw COCO captions while filtering out generic or abstract concepts. The specific instructions and negative constraints are detailed in Table 12.

## F.2. Stage 2: Knowledge-Enhanced Query Generation

For validated entities, we employ a second prompt to transform the explicit object name into an implicit reasoning query. This prompt enforces the six reasoning strategies (Logical, Functional, Cultural, Encyclopedic, Tool, Contextual) discussed in the main paper. The detailed prompt is shown in Table 13.

# G. Qualitative Case Study On KARR-Bench

In this section, we provide a comprehensive analysis of the retrieval behaviors on the KARR-Bench test set. By examining specific failure cases, we highlight how baseline methods (Full FT and LoRA) differ from our proposed SLQ in terms of reasoning depth and knowledge alignment. We discuss Text-to-Image retrieval (Figure 10) and Image-to-Text retrieval (Figure 11) respectively.

**Analysis of Text-to-Image Retrieval.** Figure 10 demonstrates the challenge of grounding implicitly defined entities. A recurrent pattern among baseline models is the inability to satisfy logical intersections between visual attributes and non-visual constraints. In the domain of biological entities, this leads to taxonomical errors. For instance, when the query specifies a "large, brown mammal" that is "domesticated" (Row 1), Full FT and LoRA retrieve wild animals like bears or highland cows, attending only to the visual adjectives while ignoring the domestication constraint. Similarly, for the "small carnivorous mammal of the weasel family" (Row 2), the baselines fail to recognize the specific biological classification, retrieving generic pets like cats, whereas SLQ correctly identifies the ferret. In the avian example (Row 6), the baselines latch onto the "bird" and water context but retrieve grey herons or geese, failing to connect the "vibrant pink feathers" and "resting on back" behavior to a flamingo.

The failure to align functional definitions with visual objects is equally pronounced in inanimate objects. In Row 3, given a query for a device to "measure mass," the baselines are misled by the kitchen context, retrieving microwaves or food processors instead of the weighing scale. For the "large pink tool used for detangling" (Row 4), the baselines exhibit shallow color matching, retrieving unrelated pink scissors or silverware, failing to link the function of "detangling" to the structure of a hairbrush. Lastly, in Row 5, the functional description of a "padded glove" for "high temperatures" is completely misinterpreted by the baselines, which retrieve raw food items or remote controls, while SLQ accurately retrieves the oven mitt used by a person. These cases collectively illustrate that SLQ possesses a superior ability to filter visual candidates based on complex logical and functional descriptions.

**Analysis of Image-to-Text Retrieval.** Figure 11 reveals the extent of hallucinations in baseline models when identifying cultural symbols and specific objects. In scenarios requiring cultural literacy, standard fine-tuning often defaults to superficial visual associations. For the Guy Fawkes mask (Row 1), baselines mistake the painted smile for a "clown" or "children's entertainer," missing the "anonymous protest" symbolism captured by SLQ. Similarly, for the "Library Way" street sign (Row 3), baselines hallucinate generic "traffic signs" or "European" contexts, whereas SLQ correctly reads the semantic cue of a "knowledge repository." In the case of the top hat (Row 4), baselines generate historically inaccurate captions about "18th-century patriots" or "religious robes," while SLQ correctly grounds the object in "19th-century formal attire."

Visual hallucinations also occur with common objects when they appear in specific contexts. In Row 2, the baselines misinterpret decorative paper kites as simply "colorful objects" or confused "wings," failing to identify them as festival decorations suspended on strings. Most strikingly, in Row 5, the baselines exhibit severe semantic drift, identifying a bunch of carrots as "cucumbers in vinegar" or a "red liquid dish," likely confused by the texture or surrounding colors. Finally, for the wrapping paper scene (Row 6), baselines hallucinate dramatic scenarios like "immobilizing a fractured limb" or "professional cooking," while SLQ accurately identifies the material used to "encase gifts." These examples underscore that SLQ maintains robust object identity and reduces hallucination compared to Full FT and LoRA.

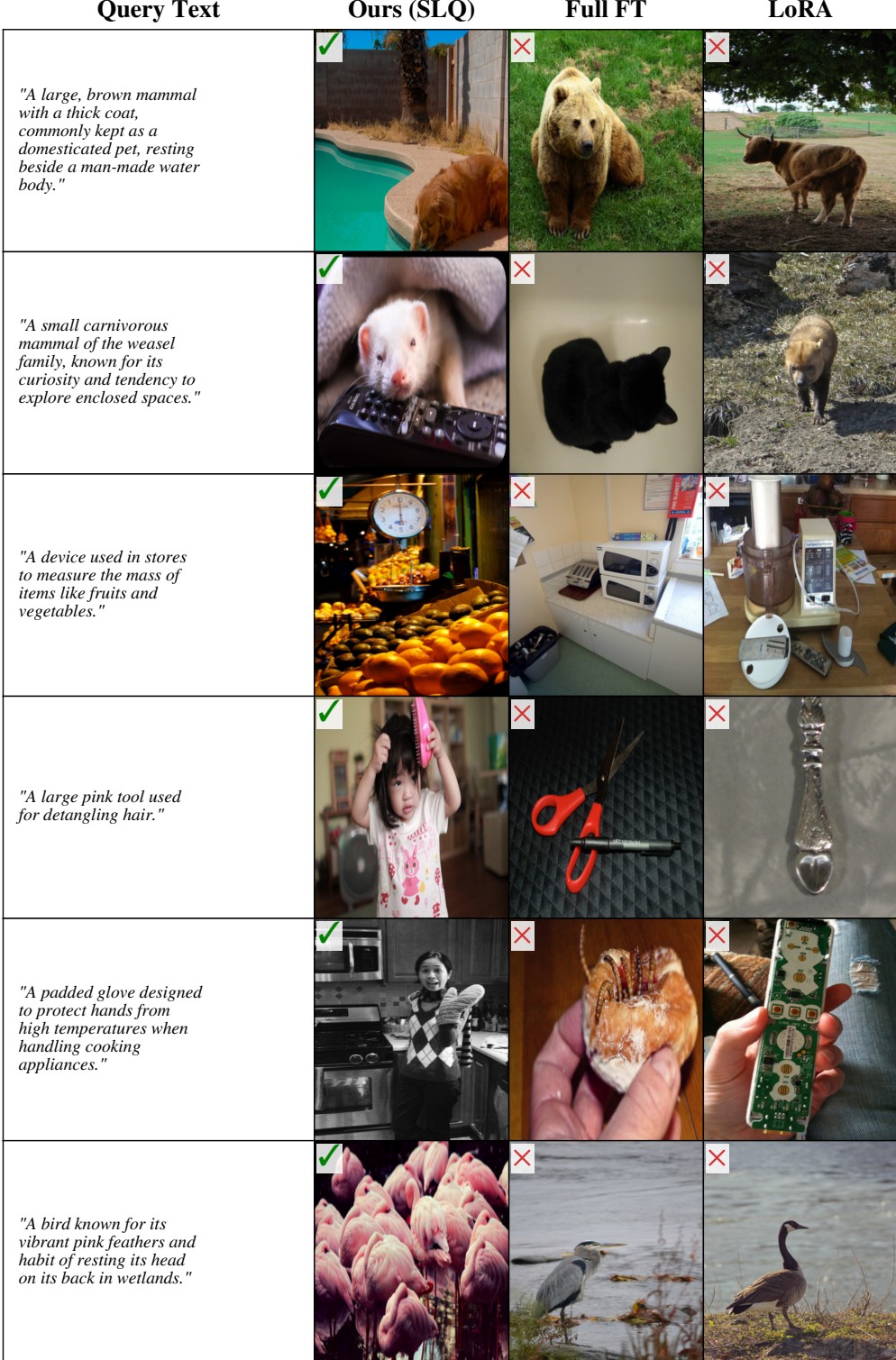

*Figure 10.* Qualitative comparison of Text-to-Image Retrieval on KARR-Bench.

| Query Image | Ours (SLQ) | Full FT | LoRA |
|---|---|---|---|
| | A symbol of anonymous protest featuring a white face with a black mustache and goatee. ✓ | A performer known for exaggerated makeup and a red nose, often associated with children's entertainment. ✗ | A performer known for exaggerated makeup and a red nose, often associated with children's entertainment. ✗ |
| | Decorative elements made of paper, suspended from a string to create a festive or ceremonial display. ✓ | A colorful object with a long tail that catches the wind and soars in the sky, held by a young girl. ✗ | A lightweight object with rainbow-striped wings, controlled by a string held by a child to soar in the air. ✗ |
| | A street named after a public knowledge repository, marked with a lion emblem in New York City. ✓ | A traffic sign indicating vehicles must proceed in a single direction. ✗ | The bilingual markers indicating directions in a European city, written in the language of Molière. ✗ |
| | A tall, cylindrical headwear historically worn by gentlemen in the 19th century, often associated with formal attire. ✓ | The attire worn by 18th-century American patriots during their fight for independence. ✗ | A person dressed in saffron robes, traditionally worn by followers of a major Eastern religion, holding a parasol and a mobile device. ✗ |
| | The orange root vegetable often used to make a vibrant sauce. ✓ | A cucumber preserved in vinegar and spices, often served as a tangy side with sandwiches. ✗ | A red liquid dish made primarily from the fruit of the Solanum lycopersicum plant, often served with grilled cheese. ✗ |
| | The material used to encase gifts during festive celebrations, often featuring holiday motifs. ✓ | The rigid material used to immobilize a fractured limb, being removed with scissors. ✗ | A professional wearing a white jacket and tall hat, typically found preparing meals in a mobile kitchen. ✗ |

*Figure 11.* Qualitative comparison of Image-to-Text Retrieval on KARR-Bench.

