# OpenReview forum: "SLQ: Bridging Modalities via Shared Latent Queries for Retrieval with Frozen MLLMs"
_ICML.cc/2026/Conference — ICML 2026 regular_

### Official Review · Reviewer_CNb5 · 2026-03-04

**Soundness:** 2
**Presentation:** 3
**Significance:** 3
**Originality:** 2
**Overall Recommendation:** 3
**Confidence:** 3

**Summary:**

This paper introduces a parameter-efficient framework that transforms frozen MLLMs into high-performance retrievers. The authors identify that existing invasive tuning methods, such as LoRA, often disrupt the pre-trained semantic manifold and lead to catastrophic forgetting. To address this, SLQ employs a small set of shared latent queries appended to both image and text sequences to aggregate multimodal context into a unified embedding space via the model's native causal attention. Additionally, the authors propose KARR-Bench, a diagnostic benchmark specifically designed to evaluate knowledge-aware reasoning in retrieval tasks. Experimental results across multiple backbones demonstrate that SLQ significantly outperforms traditional tuning strategies with orders-of-magnitude fewer trainable parameters, particularly on tasks requiring complex logical inference.

**Compliance With Llm Reviewing Policy:**

Affirmed.

**Final Justification:**

The authors have provided extensive experimental results in their rebuttal, which do address my concerns. However, I believe these experiments should be considered the original manuscript. I hope these additional results will help the authors further improve and polish their paper.

**Key Questions For Authors:**

- Limited Conceptual Novelty. While the proposed SLQ framework demonstrates strong empirical performance, its conceptual novelty is somewhat limited. The strategy of using a small set of learnable latent queries to aggregate information from a frozen backbone is a well-explored paradigm in a series of work such as CoOp[1].
- Lack of Evaluation on Recent Comprehensive Benchmarks. The empirical evaluation would be more convincing if the authors had included results on more recent and comprehensive MLLM-based retrieval benchmarks, such as MMEB[2].
- Lack of Comparison with Recent Works. In Table 1, I think it's necessary to evaluate the VLM2VEC[2] on CoCo, which constructs a critical baseline for this work. Furthermore, the authors should also evaluate the VLM2VEC[2]  on KARR-Bench.
- Missing training time comparison. The number of tunable parameters is only part of the metric to evaluate efficiency. Including the training time comparison with baselines such as LoRA, VLM2VEC would further strengthen the claim of this work.



Reference:

[1] Conditional Prompt Learning for Vision-Language Models.

[2] VLM2Vec: Training Vision-Language Models for Massive Multimodal Embedding Tasks.

**Limitations:**

The limitation is not discussed.

**Strengths And Weaknesses:**

- The motivation is clear.
- This paper is well-written and easy to follow.
- The experiments are extensive and demonstrate the effectiveness on current settings.

---

> ### Author Rebuttal · Authors · 2026-03-29
>
> We sincerely thank your thoughtful and constructive feedback. Below, we address your questions in detail.
>
> **Q1: Novelty**
>
> A1: The main contribution of multimodal prompt learning is to optimize prompt conditions, enabling existing retrieval models (e.g., CLIP) to better generalize to downstream tasks. In contrast, SLQ guides generative MLLMs, efficiently transforming them into multimodal retrieval models.
>
> - Existing methods (e.g., CLIP-based CoOp[1] / VPT[2]) prepend learnable tokens to the input sequence. Benefiting from bidirectional attention, these prompt tokens act as conditioning signals to guide feature extraction. Crucially, the final embedding is typically extracted from a separate token (e.g., [CLS]), rather than from the prompt tokens themselves. The primary goal of prompt tuning in these frameworks is to adapt the model to downstream domains.
>
> - In contrast, SLQ appends learnable tokens to the end of the sequence. Under the causal attention of decoder-only MLLMs, these tokens function purely as information aggregators. They can attend to all preceding inputs and directly aggregate full multimodal context into a compact representation that serves as the final retrieval embedding. To the best of our knowledge, SLQ is the first to show that generative MLLMs can be directly and effectively repurposed as strong multimodal retrievers.
>
> **Q2: Evaluation Results on MMEB Benchmark**
>
> A2: We add evaluation results on MMEB.
>
> Table A2: MMEB Results.
> | Model | Classification (10) | VQA (10) | Retrieval (12) | Grounding (4) | IND (20) | OOD (16) | Overall (36) |
> | :--- | :---: | :---: | :---: | :---: | :---: | :---: | :---: |
> | **No Fine-tuning on MMEB** | | | | | | | |
> | BLIP2 | 27.0 | 4.2 | 33.9 | 47.0 | 25.3 | 25.1 | 25.2 |
> | SigLIP | 40.3 | 8.4 | 31.6 | 59.5 | 32.3 | 38.0 | 34.8 |
> | E5-V  | 21.8 | 4.9 | 11.5 | 19.0 | 14.9 | 11.5 | 13.3 |
> | MagiLens | 38.8 | 8.3 | 35.4 | 26.0 | 33.1 | 23.7 | 27.8 |
> | SLQ-1B | 33.8 | 11.0 | 35.0 | 53.5 | 30.8 | 29.1 | 30.1 |
> | SLQ-2B | 33.6 | 10.7 | 39.2 | 55.4 | 32.0 | 30.6 | 31.4 |
> | SLQ-4B | 35.8 | 11.5 | 43.7 | 56.2 | 34.5 | 32.5 | 33.8 |
> | SLQ-8B | 41.1 | 10.5 | 47.6 | 59.7 | 37.8 | 35.5 | 36.8 |
> | **Fine-tuning on MMEB** | | | | | | | |
> | VLM2VEC | 61.2 | 49.9 | 67.4 | 86.1 | 67.5 | 57.1 | 62.9 |
>
> SLQ is trained only on the 118K COCO training set and not on the MMEB training data, which limits its performance. However, compared to baselines that are also not trained on MMEB, SLQ-1B outperforms E5-V-7B, demonstrating promising zero-shot generalization. As future work, we will scale up the training data to further improve competitiveness.
>
> **Q3: Comparison with VLM2VEC**
>
> A3: We evaluate the performance of VLM2VEC-Qwen2VL-7B on COCO and KARR-Bench.
>
> Table A3-1: Performance on COCO
> | Model | I→T R@1 | I→T R@5 | I→T R@10 | T→I R@1 | T→I R@5 | T→I R@10 |
> |---|---|---|---|---|---|---|
> | VLM2VEC-7B | 68.5 | 88.4 | 93.4 | 49.2 | 73.8 | 83.3 |
> | SLQ-1B | 61.2 | 84.9 | 91.8 | 48.5 | 74.3 | 83.1 |
> | SLQ-2B  | 62.7 | 84.4 | 90.7 | 50.2 | 75.7 | 83.9 |
> | SLQ-4B  | 64.3 | 85.6 | 91.2 | 50.4 | 75.2 | 83.4 |
> | SLQ-8B | 69.6 | 89.1 | 93.8 | 55.4 | 79.7 | 86.8 |
>
> Table A3-2: Performance on KARR-Bench (Recall@5)
> | Method | Image → Text | Text → Image |
> | :--- | ---: | ---: |
> | VLM2VEC-7B | 49.9 | 50.1|
> | SLQ -1B | 44.9 | 44.1 |
> | SLQ -2B | 47.3 | 47.9 |
> | SLQ -4B | 51.8 | 51.6 |
> | SLQ -8B | 58.7 | 55.7 |
>
> SLQ consistently outperforms VLM2VEC on both COCO and KARR-Bench. Notably, despite VLM2VEC being trained on large-scale data while SLQ is trained only on the COCO dataset, SLQ shows a clear advantage on KARR-Bench, where SLQ-4B even surpasses VLM2VEC-7B.
>
> **Q4: Training Time Comparison**
>
> A4: We quantitatively measure training time to demonstrate SLQ’s efficiency by reporting the time required to train for five epochs on COCO.
>
> Table A4: Training Cost Comparison
>
> | Backbone | Method | Trainable Params | Batch Size / GPU | GPU | Train Time | GPU Hours |
> | :---: | :--- | :--- | :---: | :---: | :---: | :---: |
> | InternVL3-1B | Full FT | 0.6B | 64 | 16 $\times$ H800 | 1:07:38 | 18.04 |
> | | LoRA | 4.4M | 64 | 16 $\times$ H800 | 0:42:57 | 11.43 |
> | | **SLQ** | **3.7K** | **128** | **8 $\times$ H800** | **0:29:32** | **3.94** |
> | InternVL3-8B | Full FT | 7.6B | 16 | 32 $\times$ H800 | 12:37:02 | 403.75 |
> | | LoRA | 20.2M | 16 | 32 $\times$ H800 | 4:03:53 | 130.09 |
> | | **SLQ** | **144K** | **32** | **16 $\times$ H800** | **2:25:53** | **38.90** |
>
> For the 1B model, SLQ drastically reduces the number of trainable parameters (only 3.7K) and requires merely 3.94 GPU hours, saving roughly 65% of training time compared to LoRA (11.43 GPU hours). For the 8B model, SLQ achieves an even greater time saving of 70% compared to LoRA (38.90 vs. 130.09 GPU hours). Furthermore, SLQ significantly lowers the overall hardware requirements, allowing us to halve the GPU count needed for both 1B and 8B scales.
>
> [1] Conditional Prompt Learning for Vision-Language Models.
>
> [2] Visual prompt tuning.

---

> > ### Author Rebuttal · Reviewer_CNb5 · 2026-04-03
> >
> > Thanks to the authors for their reply. My concerns have been partially addressed. However, I still believe a fair comparison with VLM2Vec and its follow-up papers is necessary.

---

> > > ### Author Response · Authors · 2026-04-04
> > >
> > > Dear Reviewer,
> > >
> > > Thank you for your valuable feedback. We agree that a fair, data-controlled comparison is necessary to validate our claims. We have fine-tuned SLQ on the same MMEB training datasets as the baselines, and include the latest concurrent works for a comprehensive evaluation.
> > >
> > > Table: Comparison with prior methods on MMEB
> > > | Model | Classification (10) | VQA (10) | Retrieval (12) | Grounding (4) | IND (20) | OOD (16) | Overall (36) |
> > > | :--- | :---: | :---: | :---: | :---: | :---: | :---: | :---: |
> > > | **No Fine-tuning on MMEB** | | | | | | | |
> > > | BLIP2 | 27.0 | 4.2 | 33.9 | 47.0 | 25.3 | 25.1 | 25.2 |
> > > | SigLIP | 40.3 | 8.4 | 31.6 | 59.5 | 32.3 | 38.0 | 34.8 |
> > > | E5-V | 21.8 | 4.9 | 11.5 | 19.0 | 14.9 | 11.5 | 13.3 |
> > > | MagiLens | 38.8 | 8.3 | 35.4 | 26.0 | 33.1 | 23.7 | 27.8 |
> > > | SLQ-1B | 33.8 | 11.0 | 35.0 | 53.5 | 30.8 | 29.1 | 30.1 |
> > > | SLQ-2B | 33.6 | 10.7 | 39.2 | 55.4 | 32.0 | 30.6 | 31.4 |
> > > | SLQ-4B | 35.8 | 11.5 | 43.7 | 56.2 | 34.5 | 32.5 | 33.8 |
> > > | SLQ-8B | 41.1 | 10.5 | 47.6 | 59.7 | 37.8 | 35.5 | 36.8 |
> > > | **Fine-tuning on MMEB** | | | | | | | |
> > > | VLM2VEC 7B [1] | **61.2** | 49.9 | 67.4 | 86.1 | 67.5 | 57.1 | 62.9 |
> > > | UniME 7B [2]  | 60.6 | 52.9 | 67.9 | 85.1 | 68.4 | 57.9 | 66.6 |
> > > | MMRet 7B [3] | 56.0 | 57.4 | 69.9 | 83.6 | 68.0 | 59.1 | 64.1 |
> > > | IDMR 8B [4]  | 58.3  | 58.6 | 68.7 | 85.6| **70.5**  | 57.9  | 64.9 |
> > > | **SLQ-1B (Ours)** | 52.5 | 49.1 | 60.2 | 78.3 | 59.8 | 54.5 | 57.3 |
> > > | **SLQ-2B (Ours)** | 54.2 | 53.3 | 62.1 | 79.5 | 62.3 | 56.0 | 59.6 |
> > > | **SLQ-4B (Ours)** | 55.7 | 58.7 | 65.4 | 84.2 | 66.5 | 58.6 | 64.5 |
> > > | **SLQ-8B (Ours)** | 60.9 | **61.2** | **70.5** | **86.6** | 69.4| **61.7** | **67.5** |
> > >
> > > 1. **Competitive performance and strong model scalability**. Under the same MMEB training mixture, SLQ-8B achieves an Overall score of 67.5, outperforming state-of-the-art methods including VLM2VEC-7B (62.9), UniME-7B (66.6), MMRet-7B (64.1), and IDMR-8B (64.9). Notably, our SLQ-4B (64.5) already surpasses both VLM2VEC-7B and MMRet-7B while utilizing significantly fewer parameters. Furthermore, the steady performance improvements from 1B to 8B demonstrate the scalability SLQ.
> > >
> > > 2. **SLQ performs strongly across diverse task categories**. On Retrieval, SLQ-8B achieves 70.5, surpassing MMRet-7B (69.9) and IDMR-8B (68.7). On Grounding, SLQ-8B reaches 86.6, which is the best among all compared models. Most notably, on VQA, SLQ-8B obtains 61.2, outperforming IDMR-8B (58.6), MMRet-7B (57.4), UniME-7B (52.9), and VLM2VEC (49.9) by a notable margin. We hypothesize that this advantage is related to keeping the MLLM backbone fully frozen. Compared with full fine-tuning or applying LoRA across the backbone, SLQ better preserves the backbone’s pre-trained visual understanding and language reasoning capabilities, which may be especially beneficial for VQA.
> > >
> > > 3. **SLQ achieves superior performance with significantly lower training cost**. Prior methods such as VLM2VEC and its follow-ups either fine-tune the full backbone or apply LoRA across all layers. In contrast, SLQ trains only a small set of latent query parameters while keeping the backbone frozen, which greatly reduces the backward computation graph. In our training setup, this leads to up to ~70% reduction in GPU training time and memory usage, while still achieving higher overall performance.
> > >
> > > We hope this comparison fully addresses your concern. We will incorporate all the suggestions and clarifications into the revised version of the paper. Thank you once again for your valuable, constructive feedback and for your consideration.
> > >
> > > [1] VLM2Vec: Training Vision-Language Models for Massive Multimodal Embedding Tasks. (ICLR 2025)
> > >
> > > [2] Breaking the Modality Barrier: Universal Embedding Learningwith Multimodal LLMs. (ACM MM 2025)
> > >
> > > [3] MegaPairs: Massive Data Synthesis for Universal Multimodal Retrieval. (ACL 2025)
> > >
> > > [4] IDMR: Towards Instance-Driven Precise Visual Correspondence in Multimodal Retrieval. (ICCV 2025)

---

### Official Review · Reviewer_q4tQ · 2026-03-12

**Soundness:** 2
**Presentation:** 3
**Significance:** 2
**Originality:** 2
**Overall Recommendation:** 3
**Confidence:** 4

**Summary:**

This paper presents SLQ, a retrieval adaptation method for frozen multimodal large language models. SLQ appends a compact set of shared learnable latent queries to both text and image token sequences, and pools the resulting query states into retrieval embeddings. In addition, the paper introduces KARR-Bench, a benchmark derived from COCO that reformulates explicit captions into knowledge- and reasoning-oriented retrieval queries. Experiments demonstrate that SLQ achieves substantially greater parameter efficiency than LoRA and fine-tuning baselines, while remaining competitive on Flickr30K and COCO and yielding more pronounced improvements on KARR-Bench. The method is validated across multiple Qwen3-VL and InternVL3 backbones, with ablation studies examining query count and embedding extraction strategies.

**Compliance With Llm Reviewing Policy:**

Affirmed.

**Key Questions For Authors:**

1. The central explanation is that SLQ preserves the pre-trained semantic manifold and therefore preserves reasoning/world knowledge better than LoRA or fine-tuning. Can you provide any direct evidence for this beyond Figure 6 and Table 3, for example performance retention on original MLLM tasks before vs after adaptation?
2. For KARR-Bench, how many human annotators were involved, what was the acceptance/rejection rate of candidate queries, and what level of agreement did you observe?

**Limitations:**

No, this paper does not contain a limitation section

**Strengths And Weaknesses:**

* Strength

1. The core idea is simple and easy to follow. The internal comparison against tuning strategies is one of the stronger parts of the paper. SLQ consistently beats the paper’s LoRA and Full FT variants while using far fewer trainable parameters.
2. The method is genuinely parameter-efficient. The trainable parameter counts are tiny relative to the backbone size
3. The paper evaluates across multiple MLLM families and scales

* Weakness

1. The paper’s main causal story is overstated relative to the evidence. The authors repeatedly attribute the gains to preserving the pre-trained semantic manifold and avoiding catastrophic forgetting. But the actual evidence for this is thin. Figure 6 on Page 8 is just a PCA projection, and Table 3 reports centroid distance. Those are at best weak correlates of alignment, not evidence that reasoning capability or world knowledge was preserved. Right now that explanation is mostly a hypothesis.
2. The novelty is meaningful but still somewhat incremental. The method is essentially continuous learned queries attached to frozen backbones for retrieval pooling, with shared parameters across modalities. That is a sensible design, but the paper tries a bit too hard to portray this as fundamentally different from prompt tuning.
3. The 'implicit CoT' claim is weakly supported. The authors suggest that query tokens elicit implicit chain-of-thought reasoning for retrieval. It may be an intuition, but it is not established.
4. The tuning-strategy comparison uses a misleading baseline name. Full FT means the vision encoder is frozen and only LLM parameters are updated. That is not what most readers mean by full fine-tuning.
5. The paper cites several recent MLLM retrieval methods in Sections 1 and 2.3, but Table 1 only compares against a subset. Since the area is moving quickly, missing strong recent baselines makes it difficult to know whether SLQ is really competitive.
6. Efficiency is argued mostly through parameter counts, which is too narrow. Figure 1 does not show real training or inference cost. If efficiency is a major point, the authors should also report the cost of time.

---

> ### Author Rebuttal · Authors · 2026-03-29
>
> We sincerely thank your thoughtful and constructive feedback. Below, we address your questions in detail.
>
> **W1 and Q1: Reasoning Capability Preservation**
>
> AW1 and A1: In Section 5.3 of the main paper, we evaluate reasoning retrieval on KARR-Bench, where SLQ consistently outperforms both LoRA and full fine-tuning. This suggests that SLQ better leverages the MLLM’s inherent reasoning abilities and world knowledge for retrieval tasks.
>
> Furthermore, we evaluate performance retention on original MLLM benchmarks using VLMToolkit, including MMMU (reasoning), RealWorldQA (world knowledge), and TextVQA (OCR).
>
> Table AW1 and A1: Performance Comparison on VQA benchmark
> | Model Scale | Method | MMMU | RealWorldQA | TextVQA |
> | :--- | :--- | :---: | :---: | :---: |
> | 1B | InternVL3  | 43.4 | 58.2 | 74.1 |
> | | SLQ (Ours) | 43.4 | 58.2 | 74.1 |
> | | LoRA | 42.1 | 56.8 | 72.6 |
> | | Full FT | 40.7 | 54.3 | 70.9 |
> | 8B | InternVL3  | 62.7 | 70.8 | 80.2 |
> | | SLQ (Ours) | 62.7 | 70.8 | 80.2 |
> | | LoRA | 60.9 | 67.2 | 78.6 |
> | | Full FT | 59.8 | 66.5 | 77.1 |
>
> As shown, SLQ keeps the MLLM backbone frozen and maintains performance consistently across VQA benchmarks. In contrast, both LoRA and full fine-tuning, when adapted for retrieval, exhibit noticeable degradation on these tasks.
>
> These results provide direct evidence that SLQ preserves the model’s original reasoning and knowledge capabilities more effectively than invasive tuning methods.
>
> **W2: Novelty**
>
> AW2:
>
> - Prompt Tuning: Existing methods (e.g., CLIP-based CoOp[1], VPT[2]) prepend learnable tokens to the input sequence. With the bidirectional attention of CLIP, these prompt tokens act as conditioning signals for feature extraction. Crucially, the final embedding is typically extracted from a separate token (e.g., [CLS]), rather than from the prompt tokens themselves. The goal of prompt tuning in these frameworks is to adapt the model to broader downstream domains.
>
> - SLQ: In contrast, SLQ appends learnable tokens to the end of the sequence. Under the causal attention of decoder-only MLLMs, these tokens function purely as information aggregators. They can attend to all preceding inputs, directly aggregating the full multimodal context into a compact representation that serves as the final retrieval embedding. To the best of our knowledge, SLQ is the first to show that generative MLLMs can be directly and effectively repurposed as strong multimodal retrievers.
>
> **W3: "Implicit CoT" claim**
>
> AW3: We acknowledge this point and will tone down the “implicit CoT” claim. However, in Figure 7 of the main paper, increasing the number of query tokens consistently improves performance on KARR-Bench, with larger gains than on standard COCO retrieval. This provides partial evidence that latent queries may act as an implicit CoT.
>
> **W4: Description of full finetune**
>
> AW4: Thank you for pointing out this error. We will correct the description.
>
> **W5: Compares with SOTA**
>
> AW5: Our work focuses on how to efficiently transform MLLMs into retrieval models. We train only on the relatively small COCO dataset and show that SLQ is more efficient than traditional LoRA and full fine-tuning. As a next step, we will scale up the training data to further improve overall performance.
>
> **W6: Training Efficiency**
>
> AW6: We quantitatively measure training time to demonstrate SLQ’s efficiency by reporting the time required to train for five epochs on COCO.
>
> Table AW6: Training Cost Comparison
>
> | Backbone | Method | Trainable Params | Batch Size / GPU | GPU | Train Time | GPU Hours |
> | :---: | :--- | :--- | :---: | :---: | :---: | :---: |
> | InternVL3-1B | Full FT | 0.6B | 64 | 16 $\times$ H800 | 1:07:38 | 18.04 |
> | | LoRA | 4.4M | 64 | 16 $\times$ H800 | 0:42:57 | 11.43 |
> | | **SLQ** | **3.7K** | **128** | **8 $\times$ H800** | **0:29:32** | **3.94** |
> | InternVL3-8B | Full FT | 7.6B | 16 | 32 $\times$ H800 | 12:37:02 | 403.75 |
> | | LoRA | 20.2M | 16 | 32 $\times$ H800 | 4:03:53 | 130.09 |
> | | **SLQ** | **144K** | **32** | **16 $\times$ H800** | **2:25:53** | **38.90** |
>
> For the 1B model, SLQ drastically reduces the number of trainable parameters (only 3.7K) and requires merely 3.94 GPU hours, saving roughly 65% of training time compared to LoRA (11.43 GPU hours). For the 8B model, SLQ achieves an even greater time saving of 70% compared to LoRA (38.90 vs. 130.09 GPU hours). Furthermore, SLQ significantly lowers the overall hardware requirements, allowing us to halve the GPU count needed for both 1B and 8B scales.
>
> **Q2: KARR-Bench Annotation Details**
>
> A2: We used GPT-5-mini (2025-08-07) for annotation and automatic filtering. From the COCO test 5000 meta data, approximately 4500 candidate samples were initially generated. Subsequently, four annotators performed cross-validation, and the final retained set contains 2,915 samples. The overall acceptance rate is approximately 60–70%.
>
> [1] Conditional Prompt Learning for Vision-Language Models.
>
> [2] Visual prompt tuning.

---

> > ### Author Rebuttal · Reviewer_q4tQ · 2026-04-08
> >
> > Thanks to the authors' detailed responses.
> >
> > 1. Regarding novelty, the proposed approach is essentially an application extension of prompt/query token-based methods in retrieval scenarios. The authors could perhaps further clarify this.
> >
> > 2. If 'retrieval model' is positioned as the core contribution, the comparison with current mainstream retrieval methods and VLM embedding models (such as Qwen-VL or Jina Embedding) remains insufficient, making its competitiveness unclear under the positioning of a 'retrieval method.' Furthermore, the evaluation solely relies on COCO and Flickr (which have been considered inadequate for validating retrieval capabilities for years), thus lacking more comprehensive validation. The authors should evaluate on datasets such as COCO-FG, Flickr-FG, Biased-COCO, Fashion-IQ, CIRR, Visual Haystacks, PhotoChat++, as well as other fine-grained, complex-query, or long-text retrieval datasets. Note that the authors should select benchmarks that highlight the specific characteristics of their proposed method for comparison. I don't mean you should evaluate on all these datasets or models.
> >
> > Overall, I believe the currently submitted version falls slightly short and lacks several key validations. Therefore, I will maintain my score.

---

### Official Review · Reviewer_Y2HJ · 2026-03-13

**Soundness:** 3
**Presentation:** 3
**Significance:** 2
**Originality:** 3
**Overall Recommendation:** 4
**Confidence:** 3

**Summary:**

The paper proposes SLQ, a parameter-efficient framework for adapting frozen multimodal large language models to dense retrieval by appending a small set of learnable shared latent queries to both text and image token sequences, then using the hidden states of these appended queries as the retrieval embedding under a symmetric contrastive objective. The motivation is that existing MLLM retrieval adaptation methods, especially full fine-tuning and LoRA, are argued to be invasive and may distort the pretrained semantic manifold that supports reasoning, while also being expensive to train. In contrast, SLQ keeps the entire backbone frozen and optimizes only the shared queries and temperature, aiming to preserve the model’s internal multimodal knowledge while eliciting a unified embedding space through causal attention. Beyond the method itself, the paper introduces KARR-Bench, a benchmark derived from COCO that replaces explicit captions with knowledge-aware, reasoning-based queries requiring functional, encyclopedic, cultural, or logical inference. Empirically, the paper reports that SLQ is more parameter efficient than LoRA and full fine-tuning, remains competitive or stronger on standard Flickr30K and COCO retrieval, and shows larger gains on KARR-Bench.

**Compliance With Llm Reviewing Policy:**

Affirmed.

**Final Justification:**

The author has resolved most of the issues through supplementary experiments and revisions to certain phrasing. Consequently, I am willing to assign a rating of "Weak Accept." However, as noted by other reviewers as well, the design of the proposed method itself suffers from a lack of novelty, which prevents me from assigning a higher score.

**Key Questions For Authors:**

* The method may rely more heavily on prompt design than the paper acknowledges. Both modalities are paired with explicit instructions such as “Summary above sentence in one word” and “Summary above image in one word.” Can you provide an ablation isolating how much of the gain comes from the shared latent queries versus the summarization prompt template itself?

* The analysis of alignment is somewhat simplistic. Using PCA visualizations and centroid distance provides an intuitive picture, but these are coarse diagnostics for representation quality. Can you show more clearly whether SLQ preserves finer semantic topology, improves calibration, or yields better local neighborhood structure, all of which would be more convincing for the paper’s manifold-preservation thesis?

* The method is mainly compared against full fine-tuning, LoRA, and a handful of retrieval systems, but there is limited discussion of stronger alternative parameter-efficient adapters or richer embedding extraction schemes beyond a linear head and separate queries. Can you choose one or two of these kinds of baselines for a more comprehensive comparison?

**Limitations:**

yes

**Strengths And Weaknesses:**

**Strengths**:
* The paper studies how to adapt strong pretrained MLLMs for retrieval without damaging the semantic structure that supports reasoning, which can broaden the application of MLLMs.
* The proposed pipeline is simple and easy to follow. Appending a small set of shared latent queries to both image and text sequences, then extracting the corresponding final hidden states as retrieval embeddings, is a clean design that leverages the backbone’s attention mechanisms naturally. It can also provide insights for investigating latent reasoning in MLLMs.
* The method updates only a tiny number of parameters relative to LoRA and full fine-tuning, yet remains competitive or stronger in retrieval performance.
* The scaling trend is interesting. The reported gap between SLQ and invasive baselines widens on larger backbones, which is consistent with the paper’s argument that stronger MLLMs contain richer but more fragile knowledge that should be preserved rather than overwritten.

**Weaknesses**:
* The paper claims that the appended queries act as an implicit chain of thought and externalize latent reasoning, but this is not directly measured. The only support is a small pilot example, plus the observation that performance improves as the number of queries increases up to 20. That does not uniquely justify the “implicit CoT” explanation over simpler alternatives such as increased representational capacity.

* KARR-Bench aims to evaluate knowledge-aware reasoning. However, except for Logical and Arithmetic, the other three strategies seem closer to re-descriptions of identity than genuinely distinct reasoning types. This makes the benchmark somewhat homogeneous and insufficiently challenging for many state-of-the-art models. This issue is further reflected in the data distribution. Functional definitions account for 54.1% of the benchmark, while cultural, encyclopedic, and logical reasoning are much less represented. Such an imbalance weakens the claim that the benchmark broadly captures knowledge-aware reasoning, since a large portion of the evaluation may largely reduce to function-centric object identification rather than diverse reasoning.

* Although clearly written, the proposed method appears largely to transfer explicit contrastive multimodal alignment into implicit latent queries. As a result, the technical novelty seems limited. In particular, the method has substantial overlap with existing ideas in cross-modal prompt learning, which makes the overall contribution appear somewhat incremental.

* The broader external significance is not fully demonstrated. The strongest case is on the proposed KARR-Bench and on zero-shot extensions in the appendix, but the paper does not yet show that SLQ generalizes to a wider range of retrieval domains such as document retrieval, long-context multimodal retrieval, or settings with substantial domain shift.

---

> ### Author Rebuttal · Authors · 2026-03-29
>
> We sincerely thank your constructive feedback and address each point below.
>
> **W1: "Implicit CoT" claim**
>
> AW1: We will tone down the "implicit CoT" claim. As shown in Figure 7, increasing query tokens consistently improves KARR-Bench (which requires reasoning) with larger gains than on COCO, suggesting latent queries may act as implicit reasoning steps.
>
> **W2: KARR-Bench**
>
> AW2: Although most samples are functional definitions, they require non-trivial reasoning. As shown in Figure 3, defining "carrot" as "a long, orange root vegetable used in salads and soups" involves semantic reasoning beyond paraphrase. We will refine the taxonomy (e.g., Tool Affordance, Biological Utility) to clarify reasoning diversity.
>
> Table AW2: SOTA comparison on KARR-Bench (Recall@5).
> | Method | Image → Text | Text → Image |
> | :--- | :---: | :---: |
> | VLM2VEC-7B | 49.9 | 50.1 |
> | SLQ-1B | 44.9 | 44.1 |
> | SLQ-2B | 47.3 | 47.9 |
> | SLQ-4B | 51.8 | 51.6 |
> | SLQ-8B | 58.7 | 55.7 |
>
> We also evaluate VLM2Vec-Qwen2VL-7B[1], which underperforms SLQ-4B, showing KARR-Bench is challenging for SOTA models.
>
> **W3: Core contribution**
>
> AW3: Multimodal prompt learning optimizes prompt conditions to adapt pre-trained retrieval models (e.g., CLIP) to downstream tasks. In contrast, SLQ is the first to demonstrate that generative MLLMs can be elegantly repurposed as powerful multimodal retrievers.
>
> **W4: Zero-Shot Generalization**
>
> AW4: We evaluate SLQ on MMEB[1] (36 tasks). Despite training only on COCO, SLQ shows strong zero-shot transfer.
>
> Table AW4: MMEB Zero-shot Results.
> | Model | Class. (10) | VQA (10) | Retrieval (12) | Grounding (4) | IND (20) | OOD (16) | Overall (36) |
> | :--- | :---: | :---: | :---: | :---: | :---: | :---: | :---: |
> | BLIP2 | 27.0 | 4.2 | 33.9 | 47.0 | 25.3 | 25.1 | 25.2 |
> | SigLIP | 40.3 | 8.4 | 31.6 | 59.5 | 32.3 | 38.0 | 34.8 |
> | E5-V-7B | 21.8 | 4.9 | 11.5 | 19.0 | 14.9 | 11.5 | 13.3 |
> | MagiLens | 38.8 | 8.3 | 35.4 | 26.0 | 33.1 | 23.7 | 27.8 |
> | SLQ-1B | 33.8 | 11.0 | 35.0 | 53.5 | 30.8 | 29.1 | 30.1 |
> | SLQ-2B | 33.6 | 10.7 | 39.2 | 55.4 | 32.0 | 30.6 | 31.4 |
> | SLQ-4B | 35.8 | 11.5 | 43.7 | 56.2 | 34.5 | 32.5 | 33.8 |
> | SLQ-8B | 41.1 | 10.5 | 47.6 | 59.7 | 37.8 | 35.5 | 36.8 |
>
> **Q1: Ablation Study on instruction**
>
> A1: All ablation studies use InternVL-1B and report Recall@5 on COCO.
>
> Removing instructions leads to a slight performance drop, indicating limited impact.
>
> Table A1: Instruction Ablation.
> | Method | Image → Text | Text → Image |
> | :--- | :---: | :---: |
> | LoRA w/o instruct | 84.2 | 72.6 |
> | LoRA w/ instruct | 84.4 | 72.9 |
> | SLQ w/o instruct | 84.5 | 73.8 |
> | SLQ w/ instruct | 84.9 | 74.3 |
>
> **Q2: Alignment Analysis**
>
> A2: Our initial evaluation follows standard practice in recent representation studies (Mind the Gap[2], E5-V[3], GME[4]). To provide a more fine-grained analysis, we additionally report Alignment and Uniformity metrics[5]. Lower Alignment indicates better positive-pair matching; lower Uniformity indicates more evenly distributed hyperspherical representations.
>
> Table A2: Alignment and Uniformity Analysis.
>
> | Method | Alignment ($\downarrow$) | Uniformity - Text ($\downarrow$) | Uniformity - Image ($\downarrow$) |
> | :--- | :---: | :---: | :---: |
> | Full FT | 0.182 | -1.15 | -1.09 |
> | LoRA | 0.155 | -1.30 | -1.26 |
> | SLQ (Ours) | 0.148 | -1.45 | -1.42 |
>
> As shown, SLQ achieves the best alignment and the lowest uniformity. The lower uniformity indicates that SLQ learns a more evenly distributed representation, supporting our manifold-preservation hypothesis. In contrast, Full FT and LoRA distort the pretrained manifold and lead to more concentrated feature subspaces.
>
> **Q3: PEFT and extraction comparison**
>
> A3: We implemented Prompt Tuning and Adapter Tuning (via HuggingFace PEFT) and a richer extraction scheme using a Transformer Block as the projection head.
>
> Table A3: Comparison with PEFT and Extraction Schemes.
> | Method | Image → Text | Text → Image |
> | :--- | :---: | :---: |
> | LoRA | 84.4 | 72.9 |
> | Prompt Tuning | 82.2 | 69.7 |
> | Adapter Tuning | 83.2 | 71.0 |
> | Head (Linear) | 78.7 | 67.4 |
> | Head (TF Block) | 78.2 | 66.7 |
> | SLQ (Ours) | 84.9 | 74.3 |
>
> Prompt Tuning performs worst due to causal masking preventing interaction with inputs. Adapter tuning underperforms LoRA, and more complex projection heads also fail to improve results, likely because they do not effectively interact with MLLM representations. This shows that increased complexity does not guarantee gains; SLQ’s native aggregation is more effective.
>
> [1] VLM2Vec: Training Vision-Language Models for Massive Multimodal Embedding Tasks.
>
> [2] Mind the Gap: Understanding the Modality Gap in Multi-modal Contrastive Representation Learning.
>
> [3] E5-v: Universal embeddings with multimodal large language models.
>
> [4] Gme: Improving universal multimodal retrieval by multimodal llms.
>
> [5] Understanding Contrastive Representation Learning through Alignment and Uniformity on the Hypersphere.

---

> > ### Author Rebuttal · Reviewer_Y2HJ · 2026-04-02
> >
> > I am very grateful to the authors for their efforts in conducting additional experiments to address my concerns. Consequently, I am willing to assign a rating of "Weak Accept." However, as other reviewers have also pointed out, the design of the proposed method itself suffers from a lack of novelty, which prevents me from assigning a higher score.

---

> > > ### Author Response · Authors · 2026-04-03
> > >
> > > We are glad to hear that our responses have addressed your concerns and questions. We sincerely appreciate you taking the time to read our rebuttal and for your positive feedback. We will incorporate all the suggestions and clarifications into the revised version of the paper. Thank you once again for your valuable, constructive feedback and for your consideration.

---

### Official Review · Reviewer_Bb8z · 2026-03-13

**Soundness:** 3
**Presentation:** 3
**Significance:** 3
**Originality:** 2
**Overall Recommendation:** 4
**Confidence:** 3

**Summary:**

This paper proposes Shared Latent Queries (SLQ), a parameter-efficient method to adapt frozen multimodal large language models (MLLMs) for image-text retrieval. The method appends a small set of learnable query tokens to the input sequence and uses their hidden states to produce embedding representations for retrieval. The model is trained using contrastive learning while keeping the backbone fully frozen.

Overall, the paper is clearly written and easy to follow. The idea is simple and intuitive, and the experimental results show that the proposed method can achieve competitive performance on standard image-text retrieval benchmarks while updating only a very small number of parameters.

However, the paper also raises several concerns. The technical novelty appears somewhat limited, as the method is conceptually close to existing prompt/query token approaches. In addition, the comparison with LoRA is somewhat difficult to interpret, and the ablation study is relatively limited given the simplicity of the proposed method.

**Compliance With Llm Reviewing Policy:**

Affirmed.

**Final Justification:**

The overall idea is very simple, not extremely novel to me. But the experiments are solid.

**Key Questions For Authors:**

1. The proposed method appears closely related to prompt tuning or query-token based approaches used in prior work. Could the authors clarify the key differences and explain why the proposed formulation should be considered substantially different from these existing techniques?

2. The results show that LoRA performs worse than SLQ despite having significantly more trainable parameters. Could the authors provide more details about the LoRA configuration (e.g., which layers are adapted, the rank used, and whether stronger settings were explored)?

3. Have the authors experimented with alternative query token placements (e.g., placing them before the input tokens) or other pooling strategies to better understand the role of the query tokens?

**Limitations:**

yes

**Strengths And Weaknesses:**

**Strengths**
1. The paper is well written and easy to follow, with a clear presentation of the motivation and method.
2. The idea is simple and intuitive, and the method appears to work reasonably well in practice.


**Weakness**
1. Limited novelty. The core idea of appending learnable query tokens to extract representations from frozen MLLMs appears conceptually close to existing techniques such as prompt tuning, prefix tuning, or query-token based representation extraction used in prior (multimodal) infomration retrieval models. The difference between the proposed approach and these existing methods is not entirely clear.

2. Comparison with LoRA is somewhat unclear. The results show that LoRA consistently underperforms the proposed method despite having significantly more trainable parameters. This outcome is somewhat surprising and raises questions about whether the training settings for LoRA are fully optimized or comparable (e.g., LoRA placement, rank, or training schedule).

3. Limited ablation study. The ablation analysis mainly studies the number of query tokens, while several important design choices are not explored. For example, it would be helpful to study the effect of query token placement (e.g., placing them before the input tokens), alternative pooling strategies, or whether modality-specific query tokens perform differently from shared ones.

---

> ### Author Rebuttal · Authors · 2026-03-29
>
> We sincerely thank your thoughtful and constructive feedback. Below, we address your questions in detail.
>
> **Q1: Novelty and Distinction from Prompt Tuning and Query-Based Methods**
>
> A1: The contribution of multimodal prompt learning is to optimize prompt conditions, enabling pre-trained retrieval models (e.g., CLIP) to generalize to downstream tasks. In contrast, SLQ is the first to demonstrate that generative MLLMs can be elegantly repurposed into powerful multimodal retrievers.  Moreover, compared to existing techniques, SLQ differs fundamentally in token placement, functional role, and architectural mechanism.
>
> 1. Token Placement and Functional Role
> - Prompt/Prefix Tuning (Prepended Conditioners): Existing methods (e.g., CLIP-based CoOp[1], VPT[2]) prepend learnable tokens to the input sequence. With the bidirectional attention of CLIP, these prompt tokens act as conditioning signals for feature extraction. Crucially, the final embedding is typically extracted from a separate token (e.g., [CLS]), rather than from the prompt tokens themselves. The goal of prompt tuning in these frameworks is to adapt the model to broader downstream domains.
> - SLQ (Appended Aggregators): In contrast, SLQ appends learnable tokens to the end of the sequence. Under the causal attention of decoder-only MLLMs, these tokens function purely as information aggregators. They can attend to all preceding inputs, directly aggregating the full multimodal context into a compact representation that serves as the final retrieval embedding. Consequently, SLQ effectively transforms a generative MLLM into a multimodal retriever.
>
> 2. Architectural Mechanism
> - Query-Based Methods: Approaches such as BLIP-2[3] introduce external modules (Q-Former), which rely on cross-attention to compress and fuse vision and language representations.
> - SLQ operates within the frozen MLLM’s native causal attention to achieve context aggregation. It introduces no additional modules, thereby preserving both the architectural integrity and the pre-trained semantic space.
>
> **Q2: LoRA configuration**
>
> A2: In the main paper, Appendix A (Table 6) lists the LoRA configuration. Following GME[5], we use r=8, α=16, dropout=0.05, applied to all linear layers of the LLM. Additionally, we conduct ablation experiments with LoRA rank ∈ {4, 8, 16, 32}.
>
> For all following ablation studies, we use the InternVL-1B and report the Recall@5 on COCO.
>
> Table A1: Ablation on LoRA ranks.
> | Method | Rank | Image→Text | Text→Image |
> | :--- | :---: | :---: | :---: |
> | LoRA | 4 | 84.5 | 72.5 |
> | LoRA | 8 | 84.4 | 72.9 |
> | LoRA | 16 | 83.9 | 72.2 |
> | LoRA | 32 | 82.1 | 71.6 |
> | SLQ (ours) | — | 84.9 | 74.3 |
>
> We observe that higher LoRA ranks do not improve performance, and full fine-tuning performs worse than LoRA, consistent with GME [4] and VLM2Vec [5]. This suggests that increasing trainable params worsen misalignment with pretrained representations due to conflict between contrastive objectives and generative pretraining. In contrast, SLQ freezes the MLLM backbone, avoids this issue, and outperforms LoRA.
>
> **Q3: Ablation study on token placement and pooling strategies**
>
> A3: In the main paper, Section 5.5 (Table 4) includes the ablation on the embedding extraction design (separate modality-specific queries).  We further provide ablations on token placement and pooling strategies.
>
> Table A3: Ablations on token placement and pooling strategies.
> | Configuration | Image→Text  | Text→Image  |
> | :--- | :---: | :---: |
> | LoRA baseline | 84.4 | 72.9 |
> | **Token Placement** | | |
> | Prepended (front) | 82.2 | 69.7 |
> | Appended (end, ours) | 84.9 | 74.3 |
> | **Query Design & Pooling** | | |
> | Modality-specific queries | 83.2 | 72.5 |
> | Last token | 84.1 | 74.1 |
> | Max pooling | 84.4 | 73.8 |
> | Mean pooling (ours) | 84.9 | 74.3 |
>
> - Ablation on Token Placement:
> Prepended queries (front) are closer in spirit to prompt tuning. Due to causal attention, prepended tokens cannot attend to subsequent tokens, creating a bottleneck, and perform worse than LoRA. In contrast, appending the queries to the end (SLQ) allows them to attend to all preceding inputs and perform global information aggregation, which is crucial for MLLM retrieval.
>
> - Ablation on Query Design and Pooling Strategies: First, shared queries outperform modality-specific queries. This suggests that parameter sharing encourages a more unified representation space across modalities. Second, among different pooling strategies (last, max, mean), mean pooling consistently performs best, as it balances information across all query positions without being dominated by any single query representation.
>
> [1] Conditional Prompt Learning for Vision-Language Models.
>
> [2] Visual prompt tuning.
>
> [3] Blip-2: Bootstrapping language-image pre-training with frozen image encoders and large language models.
>
> [4] GME: Improving Universal Multimodal Retrieval by Multimodal LLMs
>
> [5] VLM2Vec: Training Vision-Language Models for Massive Multimodal Embedding Tasks.

---

> > ### Author Rebuttal · Reviewer_Bb8z · 2026-04-02
> >
> > I will increase my score, thanks!

---

> > > ### Author Response · Authors · 2026-04-03
> > >
> > > We are pleased that our responses have addressed your questions and concerns. We truly appreciate the time you spent reviewing our rebuttal and your positive feedback. We will incorporate all the suggestions and clarifications into the revised version of the paper. Thank you again for your valuable and constructive comments, as well as your thoughtful consideration.

---

### Decision · Program_Chairs · 2026-04-30

**Decision:**

Accept (regular)

**Comment:**

This paper studies how to efficiently adapt frozen MLLMs for retrieval using shared latent queries. The authors also introduce KARR-Bench for evaluating knowledge-aware retrieval beyond surface-level matching. During rebuttal, the reviewers generally agree that the method technically sound and empirically promising, and the authors addressed most of the concerns by adding ablations and more comparisons. The main remaining concerns are that the method’s conceptual novelty is moderate to some extent, even for the positive reviewers. Besides, some of the strongest validation, especially against more comprehensive recent benchmarks and baselines, are presented during the rebuttal, then the manuscript might require a significant revision before publication. Overall, I recommend Weak accept.